# HippoTune: A Hippocampal Associative Loop–Inspired Fine-Tuning Method for Continual Learning

**Yanxi Chen**[1]    **Xiuxing Li**[1*]  **Yuyang Han**[2]    **Zhuo Wang**[1]    **Qing Li**[1]    **Ziyu Li**[1]
**Xiang Li**[3†]    **Chen Wei**[3]    **Xia Wu**[1]
[1]School of Computer Science and Technology, Beijing Institute of Technology
[2]School of Artificial Intelligence, Beijing Normal University
[3]Li Auto Inc.
{yanxi, xxl, wuxia}@bit.edu.cn

## Abstract

Studies have shown that catastrophic forgetting primarily stems from the difficulty of reactivating old memories; although parameter-efficient fine-tuning can mitigate forgetting while keeping most model parameters frozen, it still falls short in fully reawakening knowledge of prior tasks. In contrast, humans can efficiently retrieve and flexibly integrate existing experiences when learning new tasks, thereby maintaining stable performance on earlier ones. During cognition, the hippocampal EC–DG–CA3–CA1 circuit engages in multiple rounds of associative recall, and its pattern-separation and memory-completion mechanisms excel at activating historical information. Inspired by this mechanism, we propose HippoTune, a latent-space iterative retrieval strategy that embeds a query–retrieve–feedback loop within each Transformer layer. Starting from the hidden state as an initial query, the model performs a few rounds of soft key–value retrieval, projects the retrieved signals back into the query, and updates it iteratively until convergence or a preset iteration limit. Theoretically, we show this process implements a Krylov-style polynomial approximation, equivalent to a differentiable second-order preconditioner, thereby deepening retrieval in a principled way. Empirically, HippoTune outperforms classical buffer-free PEFT-CL methods by 5–8% in accuracy across three vision benchmarks, while reducing training FLOPs by 50%, effectively mitigating forgetting under tight compute constraints. Code is available at: https://github.com/yan4xi1/HippoTune.

## 1 Introduction

Deep neural networks excel under independent and identically distributed (i.i.d.) data, yet in *continual learning* (CL) settings where tasks arrive sequentially and distributions shift, they often suffer *catastrophic forgetting*: performance on earlier tasks degrades sharply when learning new ones McCloskey & Cohen (1989); Kirkpatrick et al. (2017). Classical approaches such as replay, regularization, and structural isolation often require large-scale fine-tuning of the entire network, leading to high computational and storage costs as models grow. Moreover, many such methods are largely model-agnostic and do not leverage inductive biases of prevalent architectures such as Transformers Rebuffi et al. (2017); Kirkpatrick et al. (2017); Mallya & Lazebnik (2018); Vaswani et al. (2017).

*Parameter-efficient fine-tuning* (PEFT) mitigates these costs by inserting a small number of trainable modules (e.g., adapters, LoRA modules, or prompts) into an otherwise frozen backbone, substantially reducing training overhead Houlsby et al. (2019); Hu et al. (2022); Lester et al. (2021); Li & Liang (2021). Recent PEFT-CL methods further maintain a "parameter/prompt pool" and, at inference time, retrieve and activate a subset of submodules using the sample representation as a query. However, this *single-step* retrieval can under-activate old-task memories and often requires a full backbone forward pass to extract high-level features for the query, introducing additional latency. In contrast, humans

---

*Corresponding author.
†This work was done while the author was at Li Auto. Now at Shopee.

performing previously learned tasks engage in multiple rounds of associative recall and integration, leading to richer reactivation of historical knowledge. Concretely, sparse cues can trigger multi-round recall via the hippocampal EC–DG–CA3–CA1 circuit, enabling pattern separation and completion without repeatedly reconstructing high-level semantic representations. This circuit serves as a core pathway for memory formation and retrieval in the brain: information flows from the entorhinal cortex (EC), through the dentate gyrus (DG) for pattern separation, is completed via auto-associative recurrence in CA3, and finally integrated in CA1 into a coherent memory trace Yassa & Stark (2011); Treves & Rolls (1994).

Inspired by the pattern separation, association, and integration mechanisms of the EC–DG–CA3–CA1 circuit, we propose **Latent Deliberation**, a *layer-internal, differentiable, iterative* retrieval mechanism. At each Transformer layer, we embed a light-weight associative loop: the previous layer's hidden state serves as the initial query; we perform soft key–value retrieval to activate relevant memories; the retrieved signal is linearly projected and fed back to update the query; the loop continues until convergence or a maximum number of iterations; finally, we fuse the per-iteration outputs to realize multi-level completion and integration of prior-task knowledge. Operating entirely in latent space avoids repeated construction of high-level features, and exposes practical budget controls via the maximum iteration count, a convergence threshold, and top-$k$ sparsity. This unified view also clarifies relationships to *prompt-pool* methods such as L2P Wang et al. (2022b), DualPrompt Wang et al. (2022a), and CODA-Prompt Smith et al. (2023): these can be seen as *single-depth retrieval*, whereas our method provides a *differentiable deepening of retrieval depth* to increase expressiveness and precision in memory access. See Fig. 1 for the differences between our method and the classic PEFT-CL approaches.

On the theory side, we characterize two key properties. First, near a fixed point, multi-step iteration implements a *Krylov subspace polynomial approximation* to the inverse Hessian, yielding an *implicit second-order preconditioner* for gradient propagation, achieving curvature correction in a finite number of steps without explicitly computing or storing second-order information Saad (2003); Martens & Grosse (2015). Second, we provide *convergence and stability* conditions based on step sizes and Jacobian spectral bounds, which translate into actionable choices for maximum iteration count, temperature, and entropy regularization; in effect, they operationalize the intuition that "longer deliberation/retrieval leads to better old-task performance" into verifiable and tunable optimization criteria Boyd & Vandenberghe (2004).

**We highlight four key contributions:**

- **A unified retrieval perspective for PEFT-CL.** We distill existing prompt-pool continual learning methods into a single key–value formulation, clarifying their shared trade-offs and the limits of one-shot retrieval.

- **Latent Deliberation: hippocampal-inspired iterative retrieval.** Drawing on the EC–DG–CA3–CA1 circuit, we embed a lightweight, multi-step soft lookup–and–feedback process within each Transformer layer, deepening memory activation without extra backbone passes.

- **Krylov-subspace preconditioning theory.** We prove that our finite-step loop implements a polynomial approximation to the inverse Hessian, acting as an implicit second-order preconditioner. We also derive convergence and stability criteria to guide iteration count, temperature, and regularization.

- **Strong performance at low compute cost.** On three vision benchmarks, HippoTune delivers substantial accuracy gains over one-shot PEFT-CL baselines while using only about half the training FLOPs, demonstrating efficiency under tight resource constraints.

## 2 RELATED WORK

**Continual Learning with Parameter-Efficient Fine-Tuning** Within PEFT paradigm, continual learning has evolved into a prominent direction, now marked by the convergence of *modularity*, *routing*, and *theoretical grounding*. Early methods such as L2P Wang et al. (2022b), DualPrompt Wang et al. (2022a), and CoDA-Prompt Smith et al. (2023) introduced learnable prompt pools with key–query retrieval to select modules during training and inference, mitigating forgetting without

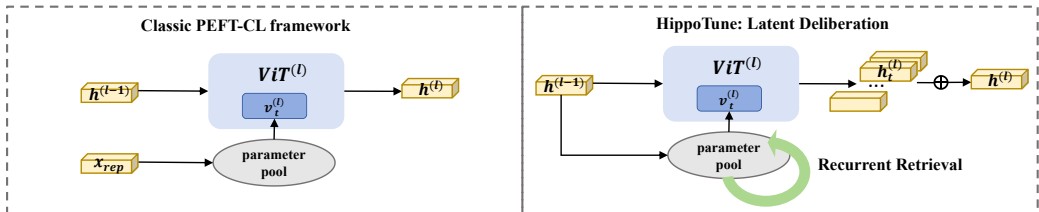

Figure 1: **Classic PEFT-CL vs. HippoTune. (Left)** Standard prompt-based continual learning retrieves a single prompt $v^{(l)}$ per ViT layer to compute $h^{(l)}$. **(Right)** HippoTune iteratively retrieves and integrates multiple prompts $\{v_1^{(l)}, \ldots, v_T^{(l)}\}$ using $h^{(l-1)}$, enabling deeper memory activation and improved retention.

relying on replay. Later approaches such as LAE Gao et al. (2023a), HiDe Zuo et al. (2023), and MoE-Adapter Yu et al. (2024) have further improved adaptability and efficiency via dynamic expansion, module merging, and expert routing. Theoretical work, including NTK analysis Doan et al. (2021) and loss landscape studies, has provided insights into how routing reduces gradient interference. However, most methods lack end-to-end optimization and seldom explore fundamental architectural principles for CL, limiting their scalability.

**Hippocampus–Neocortex Inspired Continual Learning**   Inspired by the hippocampus–neocortex interplay, continual learning research has proposed the Complementary Learning Systems (CLS) theory: the hippocampus rapidly encodes new experiences, while the neocortex gradually extracts generalized knowledge. Building on this, models like FearNet Kemker & Kanan (2018), CLS-ER Arani et al. (2022), and Triple Memory Networks Wang et al. (2021) employ short- and long-term memory modules to balance fast adaptation with long-term retention via experience replay. Key cognitive mechanisms such as hippocampal replay, pattern separation (DG), and pattern completion (CA3) have been abstracted into algorithmic strategies. Some models adopt key–value memory for associative retrieval, while GATE Liu et al. (2025) simulates gated pathways across hippocampal subregions. Despite improving the stability–plasticity trade-off, these brain-inspired methods are often architecturally complex, replay-dependent, and rarely applied under the PEFT paradigm. In this work, we propose a fine-grained emulation of hippocampal associative memory, aligned with the EC–DG–CA3–CA1 circuit. We further validate its biological plausibility and computational efficacy from both theoretical and empirical perspectives.

## 3 METHODOLOGY

We unify all PEFT modules into a shared retrieval pool and perform iterative key–value lookups and one-shot fusion at each Transformer layer, mimicking EC–DG–CA3–CA1 hippocampal loops to dynamically activate and integrate past-task knowledge. The model is trained end-to-end with classification, orthogonality, and entropy losses, using truncated BPTT to align training and inference budgets.

### 3.1 PROBLEM DEFINITION

In the continual learning (CL) setting, the model is exposed to a sequence of tasks $\{\mathcal{T}_1, \mathcal{T}_2, \ldots, \mathcal{T}_L\}$, each associated with a dataset $\mathcal{D}_t = \{(x_i, y_i)\}$. The goal is to learn a new task $\mathcal{T}_t$ while maintaining performance on previous tasks $\{\mathcal{T}_1, \ldots, \mathcal{T}_{t-1}\}$. Formally, given a model output $f(x; \Theta)$, we aim to optimize:

$$\min_{\Theta} \sum_{k=1}^{t} \mathcal{L}_{\text{cls}}^{(k)}\big(f(x; \Theta)\big), \tag{1}$$

where $\mathcal{L}_{\text{cls}}^{(k)}$ denotes the cross-entropy classification loss for task $k$.

### 3.2 PEFT-CL FRAMEWORK FORMALIZATION

In this subsection, we present a formalization of the PEFT-CL framework: we unify all lightweight modules into a shared retrieval pool, define how to compute relevance scores from a frozen backbone state, and show how to aggregate module outputs to update the model representation.

We collect all $m$ parameter-efficient modules into a single pool

$$\mathcal{V} = \{\theta^{(1)}, \ldots, \theta^{(m)}\},$$

indexed by a learnable key matrix

$$K = [k^{(1)}, \ldots, k^{(m)}]^\top \in \mathbb{R}^{m \times d}.$$

Each $\theta^{(i)}$ parameterizes a small PEFT block that takes a layer hidden state as input and produces a residual update (for example, an adapter, a prompt-induced projection, or a LoRA-style low-rank block).

Following prior works Gao et al. (2023b); He et al. (2022), we use $\phi(x; \theta^{(i)})$ as an abstract notation to unify these different PEFT modules; details of the specific form are provided in Appendix A. Here $\phi(\cdot; \theta^{(i)})$ denotes the forward mapping of the $i$-th PEFT module applied to the hidden state $x$.

Given a frozen-backbone hidden state $x \in \mathbb{R}^d$, we first compute the routing scores

$$s = \frac{x\,K^\top}{\tau}, \quad g = \mathrm{softmax}(s) \in \Delta^{m-1}, \tag{2}$$

with optional Top-$k$ truncation. Each module then emits a residual $\Delta h^{(i)} = \phi(x; \theta^{(i)}) \in \mathbb{R}^d$, which can be understood as the effect of the $i$-th PEFT module on this layer's representation. We stack all residuals as

$$\Delta H = [\Delta h^{(1)}, \ldots, \Delta h^{(m)}]^\top.$$

Starting from the current backbone state $h = x$, we conceptually update it by mixing all module outputs with the routing weights $g$:

$$h \leftarrow h + g^\top \Delta H. \tag{3}$$

In implementation, this update is realized by integrating the PEFT modules into the backbone block so that the layer directly outputs the updated $h$; the residual formulation above is an equivalent, unified view used for analysis. We provide a detailed explanation in Appendix A on how classical PEFT-CL methods correspond to this framework.

**Why this unification matters.**

1. **Query cost.** Using the model's hidden output as the retrieval query leverages rich semantic features but incurs extra computation.

2. **Retrieval depth.** All existing PEFT-CL methods perform only a single retrieval; this framework points naturally to deeper, iterative retrieval strategies.

3. **Key-gating design.** Learning and regularizing $K$, and choosing temperature, Top-$k$ or entropy penalties, determines which modules activate.

### 3.3 Latent Deliberation

At each Transformer layer, we extend the standard forward pass into a controllable dynamic process, modeled as an iterative associative loop.

We treat the hidden state from the previous layer $h^{(l-1)} \in \mathbb{R}^d$ as the initial query: $q^{(1)} = h^{(l-1)}$. Each layer maintains a learnable *key* matrix $K^{(l)} \in \mathbb{R}^{M \times d}$ and *value* matrix $V^{(l)} \in \mathbb{R}^{M \times d_v}$ to capture old-task subspaces. This is inspired by the EC–DG structure in the hippocampus, where $K^{(l)}$ acts as a guide to fixed-point memory.

At step $t$, the query $q^{(t)}$ retrieves from memory via:

$$S^{(t)} = \mathrm{softmax}\left(\frac{q^{(t)}K^{(l)\top}}{T}\right), \quad v^{(t)} = S^{(t)}V^{(l)}, \tag{4}$$

where the temperature $T > 0$ modulates retrieval sharpness. Optionally, Top-$k$ filtering can be applied to $S^{(t)}$ to retain only the most relevant memory slots. The top-$k$ hyperparameter is robust within the

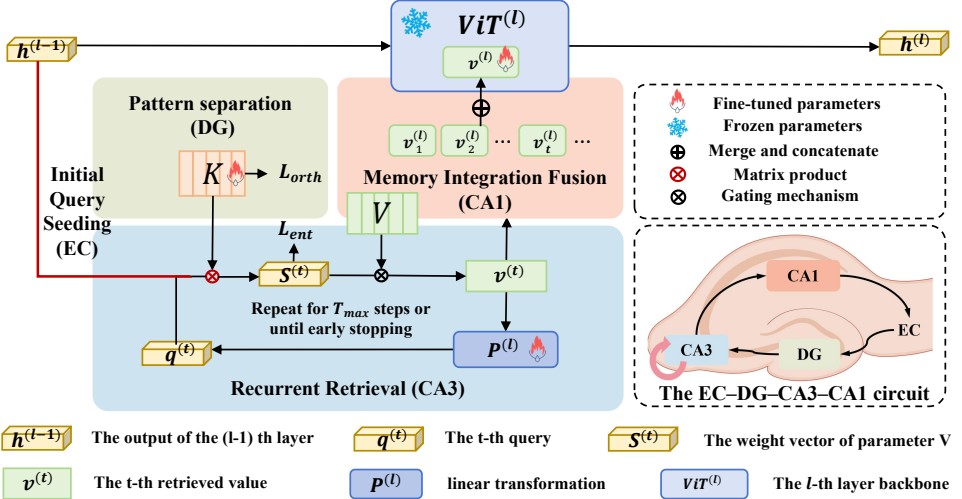

Figure 2: **A comparative illustration of HippoTune.** At each Transformer layer, use the hidden state as an initial query to iteratively perform key–value soft retrieval (with orthogonality and entropy regularization), update the query via projected residual feedback until convergence or max iterations, then fuse all retrievals into a memory-enhanced output, enabling selective multi-round activation of parameter-pool submodules.

range of 3 to 10. As this is a standard setting for prompt-based methods, we omit it from further discussion.

Then, the query is updated by incorporating the retrieved memory:

$$q^{(t+1)} = \alpha\, q^{(t)} + (1 - \alpha)\, P^{(l)}\big(v^{(t)}\big), \tag{5}$$

where $P^{(l)}$ is a layer-specific linear transformation, and $\alpha \in [0, 1]$ controls the blending. *The CA3 region features an auto-associative recurrent mechanism and can be regarded as the core of associative memory. This can be seen as a minimal abstraction of the recurrent CA3 circuit performing memory completion and state integration.* The loop terminates when either $\|v^{(t)} - v^{(t-1)}\|^2 < \varepsilon$ or $t = T_{\max}$.

To avoid repeated forward passes after each retrieval iteration, we adopt a one-shot fusion strategy, which integrates all retrieved vectors within the latent space in a unified manner. Specifically, the retrieval vector $v^{(t)}$ obtained at each iteration $t$ is concatenated along the feature dimension to form an aggregated retrieval vector:

$$V_{\mathrm{cat}} = \big[v^{(1)} \,\|\, v^{(2)} \,\|\, \cdots \,\|\, v^{(T)}\big] \,\in\, \mathbb{R}^{T d_v}, \tag{6}$$

where $\|$ denotes vector concatenation.

Next, the output of the $(l-1)$-th layer, denoted as $h^{(l-1)}$, is combined with the concatenated retrieval vector $V_{\mathrm{cat}}$ and fed into the $l$-th layer's ViT block to produce the output of layer $l$:

$$h^{(l)} = \mathrm{ViT}^{(l)}\Big(\big[h^{(l-1)} \,\|\, V_{\mathrm{cat}}\big]\Big), \tag{7}$$

where $\mathrm{ViT}^{(l)}$ represents the backbone network of the $l$-th layer. This one-shot fusion operation corresponds to the $CA1$ region in the hippocampal circuit, which is responsible for integrating the retrieved information from both $DG$ and $CA3$ and producing a complete memory representation.

Importantly, this mechanism enables explicit controllability during inference via hyperparameters such as $T_{\max}$, $\varepsilon$, and Top-$k$, allowing flexible trade-offs between retrieval quality and efficiency. The method framework is shown in Fig. 2. The pseudocode is provided in Appendix B.

### 3.4 END-TO-END TRAINING OBJECTIVE

We design a unified loss to jointly optimize task performance, retrieval sparsity, and module disentanglement:

$$\mathcal{L} = \mathcal{L}_{\text{cls}} + \lambda_{\text{orth}} \, \mathcal{L}_{\text{orth}} + \lambda_{\text{ent}} \, \mathcal{L}_{\text{ent}} \,. \tag{8}$$

- **Classification Loss** $\mathcal{L}_{\text{cls}}$: Cross-entropy loss supervising downstream performance:

$$\mathcal{L}_{\text{cls}} = -\frac{1}{N} \sum_{i=1}^{N} \sum_{c=1}^{C} y_{i,c} \log p_{i,c} \,, \tag{9}$$

where $y_{i,c}$ is the one-hot label and $p_{i,c}$ the predicted probability.

- **Orthogonality Regularization** $\mathcal{L}_{\text{orth}}$: Encourages keys $K^{(l)}$ to be orthogonal, reducing memory interference:

$$\mathcal{L}_{\text{orth}} = \sum_l \left\| K^{(l)^\top} K^{(l)} - I \right\|_F^2 \,. \tag{10}$$

- **Entropy Regularization** $\mathcal{L}_{\text{ent}}$: Controls the entropy of retrieval weights $S^{(t)}$, balancing sharpness and robustness:

$$\mathcal{L}_{\text{ent}} = -\sum_l \sum_{t=1}^{T} \sum_i S_i^{(t)} \log S_i^{(t)} \,. \tag{11}$$

The weights $\lambda_{\text{orth}}$ and $\lambda_{\text{ent}}$ balance these objectives, guiding the model towards disentangled, controllable, and generalizable behaviors.

During training, we adopt **Truncated Backpropagation Through Time (BPTT)**, propagating gradients only through the final steps of the retrieval loop. This design aligns with the dynamic budget at inference (e.g., $T_{\max}$, Top-$k$), ensuring consistency between training and deployment.

## 4 THEORETICAL ANALYSIS: MULTI-STEP RECURRENCE AND HIGHER-ORDER PRECONDITIONING

We abstract a single-layer "recurrence" as gradient descent on a smooth potential function $\phi(q)$:

$$q^{(t+1)} \;=\; q^{(t)} - \eta \, \nabla \phi\big(q^{(t)}\big), \; t = 1, 2, \ldots, T_{\max} - 1, \tag{12}$$

with step size $\eta > 0$. The outer loss depends only on the final state $q^{(T_{\max})}$, so we denote

$$g_{\text{out}} \;=\; \frac{\partial \mathcal{L}}{\partial q^{(T_{\max})}}. \tag{13}$$

**Proposition 1** (Krylov Subspace Polynomial Approximation). *Suppose $\phi$ is twice differentiable near a fixed point $q^\star$, and its Hessian $H = \nabla^2 \phi(q^\star)$ is symmetric positive definite. Further assume the step size satisfies $\rho(I - \eta H) < 1$. Define*

$$J = I - \eta H, \;\; P = \left. \frac{\partial \, (step)}{\partial \theta} \right|_{q^\star}, \;\; b_\theta = P^\top g_{\text{out}}. \tag{14}$$

*Then the leading term of the gradient w.r.t. parameters $\theta$ after $T_{\max}$ steps is*

$$\nabla_\theta \mathcal{L}_{T_{\max}} = \sum_{k=0}^{T_{\max}-1} (J^\top)^k b_\theta \;=\; \mathcal{K}_{T_{\max}}(H) \, b_\theta, \tag{15}$$

*where*

$$\mathcal{K}_{T_{\max}}(H) \;=\; \sum_{k=0}^{T_{\max}-1} (I - \eta H^\top)^k \tag{16}$$

*is the Krylov series operator. As $T_{\max} \to \infty$, the Neumann series converges and*

$$\mathcal{K}_{T_{\max}}(H) \to (\eta \, H^\top)^{-1}, \; \implies \nabla_\theta \mathcal{L}_\infty \approx H^{-1} b_\theta. \tag{17}$$

The detailed proof is provided in Appendix C.

**Corollary 1** (Effect of Finite-step Preconditioning). *For any finite $T_{\max}$, the operator $\mathcal{K}_{T_{\max}}(H)$ is a truncated polynomial approximation of $H^{-1}$ in the Krylov subspace. In practice, $T_{\max} = 2 \sim 4$ already yields effective second-order correction at only linear computational cost.*

| Method | GFLOPs | Seq-CIFAR100 | | Seq-ImageNet-R | | Seq-CUB200 | |
| --- | --- | --- | --- | --- | --- | --- | --- |
| | | Acc | AAA | Acc | AAA | Acc | AAA |
| **Classical-CL (w/ buffer)** | | | | | | | |
| LwF Li & Hoiem (2016) | 16.88 | 80.29±0.86 | 87.33±0.73 | 60.74±0.51 | 68.55±0.65 | 69.75±1.37 | 80.45±2.08 |
| DER++ Buzzega et al. (2020) | 16.88 | 84.50±1.67 | 90.16±0.61 | 54.21±0.52 | 65.26±0.58 | 77.42±0.71 | 83.61±0.09 |
| **PEFT-CL (w/o buffer)** | | | | | | | |
| L2P Wang et al. (2022b) | 35.20 | 82.76±1.17 | 88.48±0.83 | 71.26±0.44 | 76.13±0.46 | 68.39±0.46 | 78.29±0.38 |
| DualPrompt Wang et al. (2022a) | 35.38 | 85.56±0.33 | 90.33±0.33 | 68.22±0.20 | 73.81±0.39 | 66.00±0.57 | 77.92±0.50 |
| CODA-Prompt Smith et al. (2023) | 35.84 | 86.28±0.26 | 91.05±0.37 | 74.05±0.41 | 78.14±0.39 | 72.45±0.51 | 78.94±0.37 |
| LAE-PreT Gao et al. (2023a) | 35.68 | 85.25±0.66 | 89.71±0.42 | 62.81±0.48 | 69.47±0.44 | 77.48±0.79 | 85.83±0.68 |
| HiDe-Prompt Zuo et al. (2023) | 35.25 | **88.25±0.24** | **92.69±0.27** | 74.65±0.14 | 78.46±0.18 | **84.27±0.16** | **88.64±0.19** |
| **Ours (w/o buffer)** | | | | | | | |
| HippoTune (ours) | 16.92 | 87.65±0.21 | 92.07±0.25 | **74.85±0.17** | **79.92±0.22** | 81.12±0.34 | 86.63±0.41 |

Table 1: Comparison of Continual Learning Methods on Seq-CIFAR100, Seq-ImageNet-R, and Seq-CUB200 in terms of Accuracy (Acc) and Average Accuracy Across All Tasks (AAA), along with Training Time.

**Practical Guidelines**

1. **Ensure convergence:** Spectrally normalize $H$ or choose a sufficiently small $\eta$ so that $\rho(I - \eta H) < 1$.

2. **Choose $T_{max}$:** A small constant $T_{max} \approx 2$–$4$ balances second-order preconditioning with computational budget.

3. **Early stopping:** When $\|q^{(t+1)} - q^{(t)}\|$ falls below a threshold, the Krylov polynomial has effectively converged and further iterations are unnecessary.

**Conclusion.** The revised derivation eliminates the erroneous "product = series sum" step and uses the chain rule and recursive expansion to rigorously demonstrate how multi-step recurrence implicitly implements Newton or natural-gradient second-order preconditioning in the gradient.

## 5 EXPERIMENTS

### 5.1 EXPERIMENTAL SETUPS

**Benchmarks** We evaluate on three mainstream vision continual-learning benchmarks: **Seq-CIFAR100** Krizhevsky (2009); Lomonaco et al. (2021) is randomly split by class into 10 subtasks, each with 10 categories; **Seq-ImageNet-R** Boschini et al. (2022) is divided into 10 subtasks of 20 classes each; **Seq-CUB200** Wah et al. (2011); Lomonaco et al. (2021) is split into 10 subtasks, each containing 20 bird species. We conduct evaluations under the Class-Incremental Learning.

**Compared Methods** We evaluate two broad classes of methods. First, classical continual learning methods: LwF Li & Hoiem (2016) is a regularization-based approach and does not use any replay buffer, while DER++ Buzzega et al. (2020) is replay-based and in our experiments maintains a memory buffer of 1000 images. Second, buffer-free PEFT-CL methods, including L2P Wang et al. (2022b), DualPrompt Wang et al. (2022a), CODA-Prompt Smith et al. (2023), LAE-PreT Gao et al. (2023a), and HiDe-Prompt Zuo et al. (2023), freeze the backbone and train only lightweight inserted modules. We include HippoTune in the buffer-free setting and re-implement all baselines under the same backbone and training regime, using published results for LAE-PreT and HiDe-Prompt.

**Implementation Details** We adopt ViT-Base/16 as the backbone Dosovitskiy et al. (2021), freeze all non-PEFT parameters, and fine-tune only the key–value projection layers of the inserted prompt modules. Training uses the Adam optimizer Kingma & Ba (2015) with a base learning rate of 0.01, a batch size of 128, and 5 epochs per subtask. Unless otherwise noted, modules are inserted into layers 1–7 and an orthogonality regularization coefficient of $\lambda = 1$ is applied. All experiments run on NVIDIA L40S GPUs without any replay buffer, and results are averaged over three random seeds. The detailed hyperparameter settings and code link are provided in Appendix D.

| Method | GFLOPs | ImageNet-R (N = 5) | | ImageNet-R (N = 10) | | ImageNet-R (N = 20) | |
|---|---|---|---|---|---|---|---|
| | | Acc | AAA | Acc | AAA | Acc | AAA |
| Full Fine-Tuning | 16.88 | 64.92±0.87 | 75.57±0.50 | 60.57±1.06 | 72.31±1.09 | 49.95±1.31 | 65.32±0.84 |
| **PEFT-CL (w/o buffer)** | | | | | | | |
| L2P Wang et al. (2022b) | 35.20 | 73.04±0.71 | 76.94±0.41 | 71.26±0.44 | 76.13±0.46 | 68.97±0.51 | 74.16±0.32 |
| DualPrompt Wang et al. (2022a) | 35.38 | 69.99±0.57 | 72.24±0.41 | 68.22±0.20 | 73.81±0.39 | 65.23±0.45 | 71.30±0.16 |
| CODA-Prompt Smith et al. (2023) | 35.84 | 76.63±0.27 | 80.30±0.28 | 74.05±0.41 | 78.14±0.39 | 69.38±0.33 | 73.95±0.63 |
| HiDe-Prompt Zuo et al. (2023) | 35.25 | 74.77±0.25 | 78.15±0.24 | 74.65±0.14 | 78.46±0.18 | 73.59±0.19 | 77.93±0.19 |
| **Ours (w/o buffer)** | | | | | | | |
| HippoTune (ours) | 16.92 | **77.16±0.28** | **81.04±0.37** | **74.85±0.17** | **79.92±0.22** | **74.06±0.25** | **79.33±0.49** |

Table 2: Comparison of Continual Learning Methods on ImageNet-R under different task numbers (N), including GFLOPs. Results are reported in terms of final Accuracy (Acc ↑) and Average Accuracy Across All Tasks (AAA ↑).

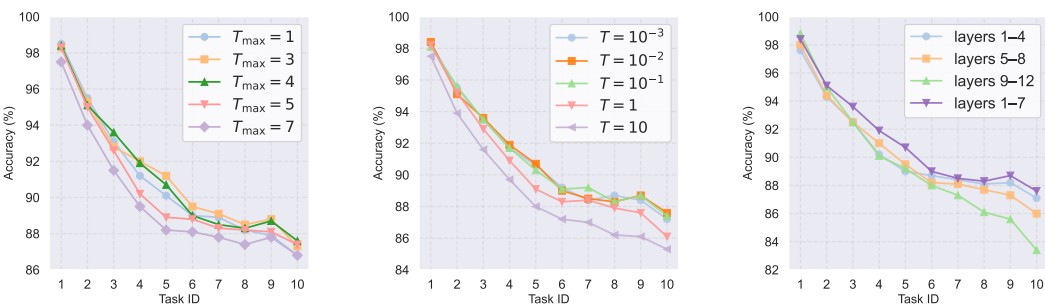

Figure 3: Further analysis on Seq-CIFAR100 for three design choices. **(Left)** Moderate max iterations (e.g., $T_{max} = 4$) balance early gains and late stability; too few/many degrade results. **(Middle)** Temperature $T$ tunes retrieval softness: mid-range ($10^{-1}$) is best; extremes underperform. **(Right)** PEFT depth: combining shallow+middle (1–7) beats shallow (1–4), middle (5–8), or deep (9–12), highlighting multi-level memory.

## 5.2 EXPERIMENTAL RESULTS

In this section, we adopt Acc and AAA as evaluation metrics. Table 1 presents the results of various methods on three benchmarks, along with their respective computational cost (GFLOPs).

- **Comparison with Classical CL Methods Using Replay Buffers** Without relying on sample replay, HippoTune achieves Acc improvements of approximately 7.4, 14.1, and 11.4 percentage points over LwF on Seq-CIFAR100, Seq-ImageNet-R, and Seq-CUB200, respectively. Compared to DER++, it yields gains of around 3.2, 20.6, and 3.7 points. In terms of AAA, HippoTune also outperforms LwF (+4.7%) and DER++ (+1.9%), clearly demonstrating that the latent-space iterative retrieval mechanism can effectively suppress forgetting without any additional memory overhead.

- **Comparison with Other PEFT-CL Methods** Compared to typical prompt-tuning methods (L2P, DualPrompt, CODA-Prompt, LAE-PreT), HippoTune achieves the highest performance on Seq-ImageNet-R with 74.85% Acc and 79.92% AAA. On Seq-CIFAR100 (87.65%/92.07%) and Seq-CUB200 (81.12%/86.63%), it also surpasses most PEFT-CL baselines, second only to HiDe-Prompt with 88.25%/92.69% and 84.27%/88.64%, respectively. The superior performance of HiDe-Prompt on these two datasets can be largely attributed to its higher computational budget and more sophisticated multi-step prompting design. Notably, HippoTune achieves better results than HiDe-Prompt on Seq-ImageNet-R (despite using only 16.92 GFLOPs compared to 35.25 GFLOPs), and delivers comparable performance on the other two benchmarks—demonstrating its efficiency and competitiveness under limited computational resources.

- **Resource Efficiency and Training Speed** With a computational cost of only 16.92 GFLOPs, approximately half that of most mainstream PEFT-CL methods (around 35 GFLOPs), HippoTune significantly improves training speed and GPU memory efficiency. Under identical hardware

settings, its training time is reduced by approximately 30% on average, confirming its practicality for scenarios with constrained computational resources.

## 5.3 ABLATION STUDY

In the ablation study, constraining the number of iterative retrieval steps to just one ($T_{\max} = 1$) leads to a notable performance drop: Acc/AAA on Seq-CIFAR100 declines from 87.65%/92.07% to 86.81%/90.63%, and on Seq-ImageNet-R from 74.85%/79.92% to 73.25%/78.62%. See Table 3 for the detailed results. This highlights the critical role of multi-step retrieval in integrating historical information and mitigating forgetting. Removing orthogonality regularization has a limited effect on Seq-CIFAR100, but results in a nearly 1.2-point drop in AAA on the more complex Seq-ImageNet-R, indicating that maintaining the diversity of retrieved vectors is especially important for leveraging prior knowledge in challenging domains. In contrast, removing entropy regularization or adopting a fusion strategy that only integrates the last-step retrieval affects overall performance by less than 0.6 points, suggesting their roles are more in stabilizing and fine-tuning the core mechanism. These findings suggest that iterative retrieval and orthogonality regularization are central to preventing catastrophic forgetting, while entropy regularization and fusion strategy can be flexibly adjusted in resource-constrained or inference-sensitive settings.

## 5.4 FURTHER ANALYSIS

**ImageNet-R under Varying Task Counts**   We split ImageNet-R into sequences of $N = 5$, 10, and 20 tasks. HippoTune consistently outperforms leading prompt-based PEFT-CL methods—gaining around 0.5–0.8 points at $N = 5$ versus CODA-Prompt and 0.2–1.5 points at $N = 10$ versus HiDe-Prompt—and even in the hardest $N = 20$ setting maintains a similar margin. Over the range $N = 5{\to}20$, its overall accuracy drops by only about 3 points, far less than typical PEFT-CL declines. Crucially, these gains come at just 16.92 GFLOPs, underscoring HippoTune's efficiency and resilience to forgetting. See Table 2 for the detailed results.

Table 3: Ablation Study: Impact of Removing Individual Components of Latent Deliberation on Seq-CIFAR100 and Seq-ImageNet-R

| Variant | Seq-CIFAR100 | | Seq-ImageNet-R | |
|---|---|---|---|---|
| | Acc | AAA | Acc | AAA |
| **Full Method** | **87.65** | **92.07** | **74.85** | **79.92** |
| Baseline | 86.28 | 90.33 | 72.93 | 78.16 |
| w/o Orthogonality Regularization | 87.32 | 91.87 | 74.09 | 78.77 |
| w/o Entropy Regularization | 87.43 | 91.30 | 74.67 | 79.55 |
| w/o Iterative Retrieval ($T_{\max} = 1$) | 86.51 | 90.63 | 72.89 | 78.10 |
| w/o Orthogonality & Entropy (Loop only) | 87.24 | 91.11 | 74.37 | 78.53 |
| w/o Iterative Retrieval & Orthogonality | 86.40 | 90.43 | 72.74 | 77.92 |
| w/o Iterative Retrieval & Entropy | 86.32 | 90.41 | 72.72 | 78.09 |
| w/o Fusion Strategy (last-step only) | 87.27 | 91.28 | 74.13 | 79.04 |
| w/o Early Stopping | 87.36 | 91.39 | 74.22 | 79.13 |

**Impact of Iteration Length**   We sweep $T_{\max} \in \{1, 3, 4, 5, 7\}$ on Seq-CIFAR100 (Fig. 3). Increasing from $T_{\max} = 1$ to 3 yields clear gains on tasks 2–6, while $T_{\max} = 4$ delivers the best overall accuracy—particularly mid-sequence—improving by about 1–2 points over $T_{\max} = 1$. Larger budgets (5 or 7) offer only marginal early-task gains and actually degrade later-task performance, suggesting that excessive iterations introduce noise or redundancy. Thus, $T_{\max} = 4$ strikes the right balance between effective memory reuse and stability.

**Accuracy Comparison Across All Tasks After Training**   Figure 4 in Appendix E.4 shows that HippoTune consistently outperforms both DER++ and DualPrompt throughout the full task sequence. In the early tasks it gains a clear lead, demonstrating its ability to recall prior knowledge immediately. This advantage persists in the mid-stage, with baseline methods trailing by a noticeable margin, and even as all methods degrade in later tasks, HippoTune's drop is the smallest. Overall, iterative retrieval both reinforces early memories and promotes steadier performance across all ten tasks.

**Impact of Temperature and Insertion Depth**   We swept the retrieval temperature $T$ from $10^{-3}$ to 10 on Seq-CIFAR100 (Fig. 3). Accuracy peaks at $T = 10^{-1}$, with mid-phase tasks (3–7) improving by 1–2 points versus $T = 10^{-3}$ and smoother convergence later. Extremes ($T = 10$ or $10^{-3}$) degrade performance, especially in mid-to-late tasks, indicating that moderate temperature best balances

knowledge sharing and task isolation. Insertion depth experiments (Fig. 3) compare placing the module in shallow (layers 1–4), middle (5–8), deep (9–12), or shallow+middle (1–7) blocks. The 1–7 configuration wins—outperforming shallow-only and middle-only by 0.5–1 point and showing smaller late-task drops than deep-only. This confirms that early-layer feature retrieval plus mid-layer memory integration yields the strongest continual-learning gains.

**Model Performance in Online Continual Learning Setting**   Our method remains highly effective even in the online setting with just one epoch of training. On seq-CIFAR100, it achieves 84.52% accuracy—less than 3% below the offline result—and surpasses the offline performance (epoch = 5) of some competing methods (see Appendix E.1).

**Effectiveness on Diverse Pre-trained Backbones**   Experiments using DINO and SAM backbones further demonstrate the strong generalization of HippoTune. As shown in Table 6 (see Appendix E.2), our method consistently achieves superior final and average accuracy across both architectures, significantly outperforming baselines like L2P and DualPrompt. Notably, it surpasses CODA-Prompt by over 4% on the SAM backbone. These results indicate that the latent iterative deliberation mechanism is architecture-agnostic and adapts well to feature distributions from diverse pre-training objectives, effectively leveraging heterogeneous representations while minimizing forgetting.

**Stability and Backward Transfer Analysis**   As shown in Table 7 (see Appendix E.3),in experiments on ImageNet-R split into 10 tasks, HippoTune outperforms standard prompt-based baselines in both accuracy and retention. While LoRA and adapter-based methods such as SD-Lora and EASE achieve high plasticity due to their architectural capacity, they suffer from significant catastrophic forgetting. In contrast, HippoTune maintains a Forgetting Measure of 4.03%, which is significantly lower than the 6% to 7% range observed in these adapter variants. This demonstrates that our method offers superior stability and effectively mitigates the interference common in high-capacity adapter approaches.

**Results in the Task-Incremental Setting**   Table 8 (see Appendix E.5) presents the comparative results under the Task-Incremental Learning (TIL) setting. HippoTune consistently outperforms the PEFT-CL baselines across both Seq-CIFAR100 and ImageNet-R benchmarks. While existing methods like CODA-Prompt already mitigate interference by conditioning on task identities, our approach further pushes the performance boundary, achieving the highest average accuracy and the lowest forgetting measures. This superiority indicates that HippoTune effectively leverages task-specific contexts to refine feature representations, ensuring robust learning of new tasks without compromising the stability of previously acquired knowledge.

## 6   CONCLUSION

We introduced HippoTune, a hippocampal-inspired continual learning method that embeds an iterative retrieval loop into each Transformer layer. By simulating the brain's multi-round associative recall and integration—combining pattern separation (DG) and completion (CA3–CA1)—HippoTune deepens memory access within PEFT frameworks without incurring repeated backbone passes. Our convergence analysis establishes a connection to Krylov-subspace second-order preconditioning, guiding choices of iteration count, temperature, and regularization. Experimentally, HippoTune delivers buffer-free gains across visual benchmarks, outperforms prompt-pool methods, and halves PEFT-CL's computational cost. Limitations include evaluation on two-level hierarchies and image classification; future work will explore deeper loops, broader modalities, and adaptive retrieval budgets to further bridge biological memory mechanisms and scalable continual learning.

ACKNOWLEDGMENTS

This work was supported by the National Science Fund for Distinguished Young Scholars of China under Grant 62325601, and the Beijing Natural Science Foundation under Grant L247011.

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

## A  DETAILED EXPLANATION OF THE PEFT-CL FRAMEWORK

### A.1  SPECIFIC METHODS WITHIN THE PEFT-CL FRAMEWORK

**L2P**  L2P maintains a fixed pool of $m$ prompt vectors and, for each example, uses the frozen backbone feature $\Phi(x)$ as the retrieval query. It computes the score vector

$$s = \frac{\Phi(x)\,K^\top}{\tau} \in \mathbb{R}^m,$$

applies a hard $\text{top-}k$ operator to obtain a sparse, one-hot–like weight $g \in \{0,1\}^m$, and concatenates the selected prompts into the input. In our PEFT-CL framework, this corresponds to $x_{\text{rep}} = \Phi(x)$, $g(s) = \text{top-}k\{s\}$ at both train and test time, and a frozen key matrix $K$, so that

$$h \;\leftarrow\; h + \text{top-}k\{s\}\,\Delta H.$$

**DualPrompt**  DualPrompt extends L2P by maintaining two disjoint pools—"general" and "domain-specific"—and by using a one-hot teacher signal $\delta_t$ during training to force selection of the correct domain prompt. At train time it sets

$$g\big(\Phi(x)K; t\big) = \delta_t,$$

and at test time uses

$$g\big(\Phi(x)K\big) = \arg\max_i \big[\Phi(x)K^\top\big]_i,$$

both yielding a one-hot weight vector. Under our formulation, $x_{\text{rep}} = \Phi(x)$, the branch outputs $\Delta H$ are mixed by these one-hot weights, and $K$ remains frozen—thus decoupling routing from the cross-entropy loss—so that

$$h \;\leftarrow\; h + g\,\Delta H.$$

**CoDA-Prompt**  CoDA-Prompt replaces hard, discrete routing with a fully differentiable soft router. Given the frozen feature $\Phi(x)$, it computes the same score $s = \Phi(x)K^\top/\tau$ but applies

$$g(s) = \text{softmax}(s) \in \Delta^{m-1}$$

to produce a dense mixture weight over all $m$ prompt branches. Importantly, both $K$ and the prompt parameters $\Delta H$ are updated via the downstream cross-entropy loss $L_{\text{CE}}$. In our unified framework this corresponds to $x_{\text{rep}} = \Phi(x)$, a trainable, differentiable routing function $g$, and joint optimization of $K$, yielding

$$h \;\leftarrow\; h + \text{softmax}(s)\,\Delta H.$$

**HiDe-Prompt**  HiDe-Prompt refines the query representation itself by learning a lightweight adapter $\widetilde{\Phi}(x)$ on top of the frozen backbone. It then reuses the DualPrompt routing strategy—one-hot teacher $\delta_t$ during training and $\arg\max$ at inference—while keeping $K$ frozen. Thus, in our PEFT-CL notation one sets $x_{\text{rep}} = \widetilde{\Phi}(x)$ and

$$g\big(\widetilde{\Phi}(x)K; t\big) = \delta_t, \quad g\big(\widetilde{\Phi}(x)K\big) = \arg\max\big(\widetilde{\Phi}(x)K\big),$$

so that

$$h \;\leftarrow\; h + g\,\Delta H,$$

with all other design choices—single retrieval, hard routing, frozen key—identical to DualPrompt.

## A.2 Instantiation of $\phi$ for common PEFT modules.

For completeness, we spell out the concrete functional form of $\phi(x; \theta^{(i)})$ for several standard PEFT blocks, where $x \in \mathbb{R}^d$ denotes the layer hidden state and $\Delta h^{(i)} = \phi(x; \theta^{(i)})$.

**Prefix Tuning.** Given a query projection $W_q^{(i)}$, a key prefix matrix $P_k^{(i)}$ and a value prefix matrix $P_v^{(i)}$, the residual contributed by the $i$-th prefix block is

$$\phi_{\text{Prefix}}(x; \theta^{(i)}) = \text{softmax}\left(x W_q^{(i)} P_k^{(i)\top}\right) P_v^{(i)}. \tag{18}$$

**Adapter.** For a bottleneck adapter with down- and up-projection matrices $W_{\text{down}}^{(i)}$ and $W_{\text{up}}^{(i)}$, we have

$$\phi_{\text{Adapter}}(x; \theta^{(i)}) = \text{ReLU}\left(x W_{\text{down}}^{(i)}\right) W_{\text{up}}^{(i)}. \tag{19}$$

**LoRA.** For a low-rank LoRA block parameterized by $W_{\text{down}}^{(i)}$ and $W_{\text{up}}^{(i)}$, the residual update is

$$\phi_{\text{LoRA}}(x; \theta^{(i)}) = x\, W_{\text{down}}^{(i)} W_{\text{up}}^{(i)}. \tag{20}$$

These instantiations are all special cases of our unified notation $\Delta h^{(i)} = \phi(x; \theta^{(i)})$ used in the main text.

## B Pseudocode

---
**Algorithm 1** Latent Deliberation: Iterative Retrieval and Integration

---
**Require:** Backbone depth $L$, max iterations $T_{\max}$, tolerance $\varepsilon$, temperature $T$, blend factor $\alpha$, (optional) Top-$k$
1: **for** $l = 1$ **to** $L$ **do**
2:     **Input:** previous hidden state $h^{(l-1)} \in \mathbb{R}^d$
3:     Initialize query: $q^{(1)} \leftarrow h^{(l-1)}$
4:     Initialize empty list $\mathcal{V} \leftarrow []$
5:     **for** $t = 1$ **to** $T_{\max}$ **do**
6:         Compute retrieval weights: $S^{(t)} \leftarrow \text{softmax}\left(q^{(t)}(K^{(l)})^\top / T\right)$
7:         **if** Top-$k$ enabled **then**
8:             Keep only top-$k$ entries of $S^{(t)}$, zero out others
9:         **end if**
10:        Retrieve memory: $v^{(t)} \leftarrow S^{(t)} V^{(l)}$
11:        Append $v^{(t)}$ to $\mathcal{V}$
12:        Update query: $q^{(t+1)} \leftarrow \alpha\, q^{(t)} + (1 - \alpha)\, P^{(l)}\left(v^{(t)}\right)$
13:        **if** $t > 1$ **and** $\|v^{(t)} - v^{(t-1)}\|^2 < \varepsilon$ **then**
14:            **break**
15:        **end if**
16:     **end for**
17:     **One-shot fusion:** $V_{\text{cat}} \leftarrow \text{concat}(\mathcal{V}) \in \mathbb{R}^{T_{\max} d_v}$
18:     Compute layer output: $h^{(l)} \leftarrow \text{ViT}^{(l)}\left([\, h^{(l-1)} \| V_{\text{cat}} \,]\right)$
19: **end for**

---

## C Proof of Proposition 1

*Proof of Proposition 1.* We regard one gradient-descent "inner" step as a map

$$F(q; \theta) = q - \eta \nabla \phi(q), \qquad q^{(t+1)} = F\left(q^{(t)}; \theta\right),$$

where $\theta$ denotes *outer* parameters that may affect the step only through some auxiliary operation (e.g. the retrieval projection in our Latent Deliberation loop). Let $q^\star$ be a fixed point of the inner dynamics so that $\nabla\phi(q^\star) = 0$. Linearising $F$ at $(q^\star, \theta)$ gives the *state Jacobian* $J = \left.\frac{\partial F}{\partial q}\right|_{q^\star} = I - \eta\,\nabla^2\phi(q^\star) = I - \eta H$ and the *parameter Jacobian* $P = \left.\frac{\partial F}{\partial \theta}\right|_{q^\star}$.

**Step 1: Recursion on sensitivities.** Differentiating the inner recurrence w.r.t. $\theta$ yields

$$\frac{\partial q^{(t+1)}}{\partial \theta} = J\,\frac{\partial q^{(t)}}{\partial \theta} + P, \qquad t = 0,\ldots,T_{\max} - 1,$$

with the base term $\frac{\partial q^{(0)}}{\partial \theta} = 0$ because the initial hidden state is taken as constant for the outer optimisation. Solving the first-order, non-homogeneous linear recursion gives

$$\frac{\partial q^{(T_{\max})}}{\partial \theta} = \sum_{k=0}^{T_{\max}-1} J^k\, P.$$

**Step 2: Chain rule for the outer loss.** Applying the chain rule,

$$\nabla_\theta \mathcal{L}_{T_{\max}} = \left(\frac{\partial q^{(T_{\max})}}{\partial \theta}\right)^\top g_{\text{out}} = \sum_{k=0}^{T_{\max}-1} (J^\top)^k P^\top g_{\text{out}} = \sum_{k=0}^{T_{\max}-1} (J^\top)^k b_\theta,$$

where $b_\theta := P^\top g_{\text{out}}$. This is exactly the Krylov series operator $\mathcal{K}_{T_{\max}}(H) = \sum_{k=0}^{T_{\max}-1}(I - \eta H^\top)^k$ acting on $b_\theta$.

**Step 3: Convergence to the Neumann–series inverse.** Because $H$ is symmetric positive definite and $0 < \eta < 2/\lambda_{\max}(H)$, the spectral radius $\rho(J) = \rho(I - \eta H) < 1$. Hence the Neumann series converges:

$$\lim_{T_{\max}\to\infty} \mathcal{K}_{T_{\max}}(H) = (I - J^\top)^{-1} = \left(\eta\,H^\top\right)^{-1}.$$

Taking the limit in the gradient expression gives $\nabla_\theta \mathcal{L}_\infty = H^{-1} b_\theta$, which corresponds to exact second-order (Newton–style) preconditioning.

**Step 4: Finite-step interpretation.** For any finite $T_{\max}$, $\mathcal{K}_{T_{\max}}(H)$ is a degree-$(T_{\max}-1)$ polynomial that approximates $H^{-1}$ in the Krylov subspace $\text{span}\{b_\theta, H^\top b_\theta, \ldots, (H^\top)^{T_{\max}-1}b_\theta\}$. Because the error decays geometrically in $\rho(J)$, practitioners often find $T_{\max} = 2$–$4$ iterations already provide most of the curvature correction at only linear cost.

This completes the proof. $\qquad\square$

## D   EVALUATION METRICS AND HYPERPARAMETERS

Our code is available at https://github.com/TODO-YOUR-REAL-REPO/HippoTune and also included in the supplementary material archive.

**Accuracy (Acc)**   For any evaluation point $i$ and task $j$, the *accuracy* is simply

$$\text{Acc}(i,j) = a_{i,j},$$

i.e. the test accuracy on task $j$ immediately after learning task $i$.

**Average Accuracy Across Tasks (AAA)**   At the end of task $i$, the *average accuracy* across all seen tasks is

$$\text{AAA}(i) = \frac{1}{i}\sum_{j=1}^{i} a_{i,j}.$$

In particular, $\text{AAA}(T)$ summarizes overall performance after the final task.

**Forgetting Measure (FM)**  For each earlier task $j < i$, its *forgetting* after learning up to task $i$ is

$$f_j^i = \max_{1 \le l < i} a_{l,j} - a_{i,j},$$

i.e. the largest drop from its best-seen accuracy to its current accuracy. The *average forgetting* at the end of all tasks is then

$$\text{FM} = \frac{1}{T-1} \sum_{j=1}^{T-1} f_j^T.$$

A smaller FM indicates less catastrophic forgetting.

We now introduce the hyperparameters one by one, including key hyperparameters of the ViT backbone as well as those specific to our HippoTune training procedure.

**learning_rate**  Initial learning rate used by the optimizer.

**batch_size**  Number of samples processed in each training batch.

**epoch**  Number of full passes over the training dataset.

**pretrained**  Whether to load pretrained backbone weights (1: yes; 0: no).

**clip_grad**  Maximum norm for gradient clipping to prevent exploding gradients.

**size**  Number of prompt vectors in the prompt pool.

**length**  Length of each prompt vector (in tokens).

**initializer**  Initialization scheme for prompt values (e.g. `uniform` for uniform distribution).

**prompt_key_init**  Initialization scheme for prompt keys (e.g. `uniform`).

**batchwise_prompt**  Whether to share the same prompt across the entire batch (1: yes; 0: per-example).

**global_pool**  Pooling method over ViT outputs: `token` (use class token) or `avg` (global average pooling).

**head_type**  Input to the classification head: `token`, `gap` (global average pooling), `prompt`, or `token+prompt`.

**freeze**  List of backbone submodules to freeze during training, e.g. `[blocks, patch_embed, cls_token, norm, pos_embed]`.

**$\lambda_{\text{orth}}$**  Weight of the orthogonality regularization term on prompt keys.

**layer_idx**  Indices of Transformer layers where prompting/retrieval is applied, e.g. `[0,1,2,3,4,5,6]`.

**$T$**  Softmax temperature for computing attention weights over keys.

**delib_steps ($T_{\max}$)**  Maximum number of latent deliberation (iterative retrieval) steps.

**$\alpha$**  Query blending ratio in each iteration: $q_{t+1} = \alpha \, q_t + (1 - \alpha) \, P(v_t)$.

**eps_stop**  Early-stop threshold on $\|v_t - v_{t-1}\|$ for terminating latent deliberation.

**topk**  Number of top keys to retain for sparse retrieval; `None` means full softmax.

**fuse**  Method to combine outputs across steps: `mean` (average) or `last`.

**$\lambda_{\text{ent}}$**  Weight for entropy regularization on the retrieval distribution.

| Dataset | pretrained | clip_grad | size | length | initializer | prompt_key_init | batchwise_prompt | lr | batch_size | epochs |
|---------|-----------|-----------|------|--------|-------------|-----------------|------------------|-----|-----------|--------|
| Seq-CIFAR100 | 1 | 1.0 | 30 | 10 | uniform | uniform | 1 | 0.001 | 128 | 5 |
| Seq-ImageNet-R | 1 | 1.0 | 30 | 10 | uniform | uniform | 1 | 0.001 | 128 | 5 |
| Seq-CUB200 | 1 | 1.0 | 30 | 10 | uniform | uniform | 1 | 0.001 | 128 | 5 |

(a) Basic Training Hyperparameters

| Dataset | global_pool | head_type | freeze |
|---------|-------------|-----------|--------|
| Seq-CIFAR100 | token | token | [blocks, patch_embed, cls_token, norm, pos_embed] |
| Seq-ImageNet-R | token | token | [blocks, patch_embed, cls_token, norm, pos_embed] |
| Seq-CUB200 | token | token | [blocks, patch_embed, cls_token, norm, pos_embed] |

(b) ViT-related Hyperparameters

| Dataset | $\lambda_{orth}$ | layer_idx | $T$ | delib_steps($T_{max}$) | $\alpha$ | eps_stop | topk | fuse | $\lambda_{ent}$ |
|---------|-----------|-----------|-----|------------------------|----------|----------|------|------|----------|
| Seq-CIFAR100 | 1.0 | [0,1,2,3,4,5,6] | 0.01 | 4 | 0.2 | 1e-5 | 5 | mean | 1 |
| Seq-ImageNet-R | 1.0 | [0,1,2,3,4,5,6] | 0.01 | 4 | 0.2 | 1e-5 | 5 | mean | 1 |
| Seq-CUB200 | 1.0 | [0,1,2,3,4,5,6,7,8] | 0.01 | 4 | 0.2 | 1e-5 | 5 | mean | 1 |

(c) Hyperparameter Settings for Multi-Key Retrieval and Latent Deliberation

# E    ADDITIONAL EXPERIMENTS

## E.1    ONLINE CONTINUAL LEARNING

Our method remains highly effective in the strictly online setting with a single data pass: on Seq-CIFAR100, **HippoTune-PreT** achieves **84.52%** $\pm$ **0.23** Acc, **89.09%** $\pm$ **0.21** AAA, and **7.48** $\pm$ **0.17** FM—only $\sim$2.8% below its offline (5-epoch) result and clearly outperforming buffer-free baselines such as L2P (75.38% $\pm$ 1.05 Acc) and DualPrompt (80.89% $\pm$ 0.58 Acc) (Table 5). On Seq-CUB200, it attains **65.99%** $\pm$ **0.24** Acc, **74.52%** $\pm$ **0.64** AAA, and **3.55** $\pm$ **0.35** FM, surpassing CODA-Prompt's 62.63% $\pm$ 0.34 Acc. The efficient variant **HippoTune-PreT-E** also yields strong performance (84.07% $\pm$ 0.28 Acc, 88.62% $\pm$ 0.36 AAA, 8.07 $\pm$ 0.15 FM on Seq-CIFAR100; 64.69% $\pm$ 0.53 Acc, 72.94% $\pm$ 0.48 AAA, 4.19 $\pm$ 0.32 FM on Seq-CUB200). Moreover, a sharing-favored prompt allocation ($T = 1$) outperforms an isolation-favored one ($T = 0.01$) by $\sim$0.8% on Seq-CIFAR100 and $\sim$1.3% on Seq-CUB200, indicating that emphasizing shared knowledge significantly aids convergence in the challenging one-pass online regime.

## E.2    EFFECTIVENESS ON DIVERSE PRE-TRAINED BACKBONES

Experiments using DINO and SAM backbones further demonstrate the strong generalization of HippoTune. As shown in Table 6, our method consistently achieves superior final and average accuracy across both architectures, significantly outperforming baselines like L2P and DualPrompt. Notably, it surpasses CODA-Prompt by over 4% on the SAM backbone. These results indicate that the latent iterative deliberation mechanism is architecture-agnostic and adapts well to feature distributions from diverse pre-training objectives, effectively leveraging heterogeneous representations while minimizing forgetting.

## E.3    STABILITY AND BACKWARD TRANSFER ANALYSIS

As shown in Table 7, in experiments on ImageNet-R split into 10 tasks, HippoTune outperforms standard prompt-based baselines in both accuracy and retention. While LoRA and adapter-based methods such as SD-Lora and EASE achieve high plasticity due to their architectural capacity, they suffer from significant catastrophic forgetting. In contrast, HippoTune maintains a Forgetting Measure of 4.03%, which is significantly lower than the 6% to 7% range observed in these adapter variants. This demonstrates that our method offers superior stability and effectively mitigates the interference common in high-capacity adapter approaches.

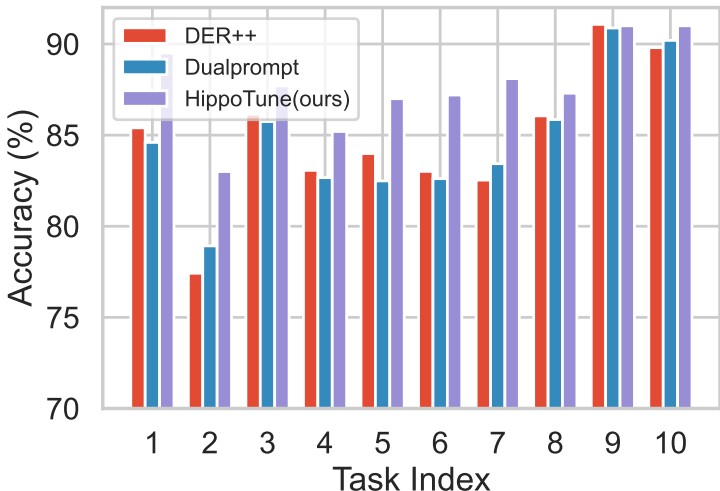

Figure 4: **Per-task accuracy comparison across 10 sequential tasks.** We report the task-wise Accuracy (%) of DER++, Dualprompt, and our HippoTune method on Seq-CIFAR100. HippoTune consistently outperforms the baselines, especially on early and late tasks, demonstrating its effective latent deliberation mechanism.

### E.4 Accuracy Comparison Across All Tasks After Training

We provide a detailed visualization of the accuracy evolution across all tasks to evaluate the stability of the model throughout the training process. As shown in Figure 4, HippoTune consistently maintains a significant performance advantage over the baseline methods across the entire sequence.

### E.5 Results in the Task-Incremental Setting

Table 8 presents the comparative results under the Task-Incremental Learning (TIL) setting. Hippo-Tune consistently outperforms the PEFT-CL baselines across both Seq-CIFAR100 and ImageNet-R benchmarks. While existing methods like CODA-Prompt already mitigate interference by conditioning on task identities, our approach further pushes the performance boundary, achieving the highest average accuracy and the lowest forgetting measures. This superiority indicates that HippoTune effectively leverages task-specific contexts to refine feature representations, ensuring robust learning of new tasks without compromising the stability of previously acquired knowledge.

Table 5: Comparison of Continual Learning Methods on Seq-CIFAR100 and Seq-CUB200 in Terms of Accuracy, Average Accuracy Across All Tasks (AAA), and Forgetting Measure (FM).

| Method | Seq-CIFAR100 | | | Seq-CUB200 | | |
|---|---|---|---|---|---|---|
| | Acc | AAA | FM | Acc | AAA | FM |
| **PEFT-CL (w/o buffer)** | | | | | | |
| L2P | 75.38±1.05 | 84.38±0.58 | 10.17±0.62 | 60.78±0.42 | 69.21±0.46 | 5.37±0.23 |
| DualPrompt | 80.89±0.58 | 86.74±0.37 | 10.32±0.55 | 62.79±0.27 | 71.25±0.53 | 4.87±0.25 |
| CODA-Prompt | 82.68±0.39 | 88.01±0.46 | 9.96±0.53 | 62.63±0.34 | 71.63±0.41 | 4.79±0.52 |
| **Ours (w/o buffer)** | | | | | | |
| **HippoTune** | **84.52±0.23** | **89.09±0.21** | **7.48±0.17** | **65.99±0.24** | **74.52±0.64** | **3.55±0.35** |

Table 6: Performance comparison of PEFT-CL methods and HippoTune on ImageNet-R (N=10) with DINO and SAM backbones (metrics: Acc, AAA, FM).

| Method | ImageNet-R (N=10, Dino) | | | ImageNet-R (N=10, SAM) | | |
|---|---|---|---|---|---|---|
| | Acc | AAA | FM | Acc | AAA | FM |
| **PEFT-CL** | | | | | | |
| L2P | 51.12 | 62.68 | **1.34** | 56.93 | 59.38 | 6.46 |
| DualPrompt | 60.18 | 66.91 | 4.40 | 65.13 | 70.05 | 5.29 |
| CODA-Prompt | 63.74 | 69.07 | 5.43 | 66.61 | 71.63 | 5.72 |
| **Ours** | | | | | | |
| **HippoTune** | **65.62** | **70.29** | 3.78 | **70.92** | **75.54** | **5.16** |

Table 7: Forgetting rates (FM) and backward transfer (bwt).

| Metric | l2p | dualprompt | codaprompt | sdlora | ease | ranpac | hippotune |
|---|---|---|---|---|---|---|---|
| FM | 4.11 | 5.18 | 5.39 | 7.34 | 6.37 | 4.62 | 4.03 |
| bwt | -4.03 | -5.18 | -5.26 | -7.24 | -6.21 | -4.56 | -3.94 |

Table 8: Task-Incremental Learning (TIL) performance comparison on Seq-CIFAR100 and ImageNet-R. We report Average Accuracy (Acc), Average After Accuracy (AAA), and Forgetting Measure (FM). Best results are highlighted in **bold**.

| Method | Seq-CIFAR100 | | | ImageNet-R | | |
|---|---|---|---|---|---|---|
| | Acc | AAA | FM | Acc | AAA | FM |
| **PEFT-CL** | | | | | | |
| L2P | 96.84 | 97.04 | 0.57 | 89.45 | 89.24 | 0.53 |
| DualPrompt | 97.25 | 97.52 | 0.62 | 87.72 | 88.21 | 0.71 |
| CODA-Prompt | 97.83 | 98.04 | 0.50 | 89.57 | 89.66 | 0.52 |
| **Ours** | | | | | | |
| **HippoTune** | **98.54** | **98.68** | **0.44** | **90.43** | **90.41** | **0.48** |

# F    THE USE OF LARGE LANGUAGE MODELS (LLMS)

In this paper, large language models (LLMs) were used only as general-purpose assistive tools for language polishing, improving writing structure, and retrieving and organizing references. LLMs did not contribute to research ideation, method design, experiment execution, or result analysis, and thus do not constitute a substantive scholarly contribution.

