# OpenReview forum: "HippoTune: A Hippocampal Associative Loop–Inspired Fine-Tuning Method for Continual Learning"
_ICLR.cc/2026/Conference — ICLR 2026 Poster_

### Official Review · Reviewer_r3mR · 2025-10-27

**Soundness:** 4
**Presentation:** 4
**Contribution:** 4
**Rating:** 8
**Confidence:** 4

**Summary:**

Recently, several continual learning (CL) methods have emerged that utilise pre-trained models through parameter-efficient fine-tuning. One category of these methods is based on soft prompts, which involve training a set of prompts to guide the representations of each input. Typically, these methods select prompts based on a feature vector of the input, computed via an additional forward pass through the model. In this work, the authors address the inefficiency of this process by proposing a new method for selecting prompts for each layer of the model, based on the representations from the previous layer. This approach reduces the number of FLOPs during training by eliminating the need for an initial forward pass to identify prompts. Inspired by the hippocampus, they introduce HippoTune, an iterative retrieval strategy designed for transformer layers. The paper includes results presented across three visual benchmarks, along with a comprehensive ablation analysis.

**Strengths:**

- The paper effectively highlights a limitation of several CL methods that use soft prompts: they perform a double forward pass per example (prompt selection and classification). Using the representation of the previous layer for prompt selection is an interesting idea that has been used in other areas, but it is presented in a novel way in the CL scenario.
- Performing multiple retrieval steps is an interesting idea, showing theoretically and empirically that it works.
- A comprehensive ablation study of the components of the proposed method is conducted.

**Weaknesses:**

- The paper discusses a framework for PEFT-CL methods, but it applies only to methods that use soft prompts. LoRA-based methods do not necessarily include a retrieval component, so they cannot be extended to the framework. It would be good to make it clear that the proposed framework applies only to methods that use a key-value selection of prompts.
    - Also, it is not clear why the proposed framework is necessary. I understand that it helps to explain the method better, but it does not necessarily help to understand the other methods better.
- There are many hyperparameters to search over (T_max, loss coefficients, T, plus the common ones for this type of method). This can increase the method's complexity.

**Questions:**

- By concatenating the values of "v" from each iteration (Eq. 6), a better representation of what is retrieved is achieved. Although the proposal achieves fewer FLOPs, this concatenation increases the batch size, since each batch element adds several elements to the sequence, resulting in a larger batch size than previous methods.
    - What percentage of the batch is used by these vectors? How much can this affect scenarios where GPU size may be an issue?
- Despite showing better results with multiple iterations (T_max >1), can you provide insight into why this improves performance?
    - Is it related to increasing the concatenated values to v?
- One problem with prompt-based methods is the low generalisation of vectors to datasets outside the distribution of the pre-training set.
    - Were experiments on datasets where these types of methods fail conducted?
    - It would be interesting to see how the proposed method would perform.

---

> ### Author Response · Authors · 2025-11-23
> **To Reviewer r3mR (Part 1)**
>
> We sincerely appreciate your valuable comments. Please find our point-by-point responses below. For your convenience, all revisions in the paper are highlighted in blue. Please let us know if you have any further questions.
>
> **Q2: Why multiple iterations work & W1: How our framework aids understanding?**
>
> > **(1) Intuitive Explanation:**
> >
> > Single-step retrieval is often inaccurate or incomplete, while multi-step iteration refines the result. Our PEFT-CL framework reveals that the key difference between methods lies in $S^{(T)}$ in Eq. 4 (the weight of the retrieved prompt). This helps us identify that memory retrieval quality is the core performance driver in other methods. It also highlights their limitations, such as the failure to fully retrieve the most appropriate memories. The significant performance gap across existing methods confirms that optimizing retrieval quality is crucial. Interestingly, this iterative process acts like an implicit Chain-of-Thought (CoT): the inferred prompt serves as input for the next step, recursively refining the memory until it is sufficiently complete.
> >
> > **(2) Theoretical Explanation:**
> >
> > In Section 4, we show that unrolling the inner loop for $T_{max}$ steps expresses the gradient for $\theta$ as a Krylov subspace polynomial on the local Hessian $H$ (Eq. 15 & 16). As $T_{max}$ increases, this approximates $H^{-1}$, creating an implicit second-order preconditioning. In other words, the iteration modifies the optimization geometry and introduces curvature correction without explicitly calculating the Hessian. This stabilizes gradients and improves alignment, which is similar to gradient-based continual learning methods that use the Hessian to balance "old" and "new" task directions.
> >
> > **(3) Is it just Concatenation?**
> >
> > Concatenation contributes to performance improvements, but it is not the decisive factor. First, the "Last-step Only" experiment in Table 3 demonstrates that using only the final retrieved prompt results in accuracy drops of 0.4% and 0.7% compared to the full method on the two datasets, respectively. Second, we explored alternative composition strategies, such as merging the retrieved prompt from each step directly with the backbone to output $h^{(l)}$ as the next query. Although this approach outperformed our current method by approximately 0.5%, we discarded it because it significantly increases training time.
> > Moreover, if the gains were solely attributable to increased dimensionality, performance should improve monotonically as $T_{max}$ increases. However, experiments with $T_{max}=7$ indicate that concatenating an excessive number of prompts actually degrades performance. We therefore conclude that both memory accuracy and completeness are critical, and a 3-4 step iterative retrieval represents the optimal balance between these two factors.
>
> **Q1: GPU Memory Overhead**
>
> > **(1) Percentage of Retrieved Prompts in Input Sequence**
> >
> > The standard input sequence length for ViT-Base/16 is $(224/16)^2 + 1 (\text{CLS}) = 197$ tokens. HippoTune adds a length of $T_{max}=4$. Assuming each $v$ has a length of 10 (refer to Table 3), this adds $4 \times 10 = 40$ tokens. Proportion: $40 / (197+40) \approx 16.9\%$.
> >
> > **(2) Addressing Concerns on GPU Memory Capacity**
> >
> > To directly address concerns regarding GPU capacity, we measured the actual peak VRAM usage during training (Batch Size=128, ViT-B/16).
> > - **Without Recursive Retrieval:** Memory usage $\approx 26.23$ GB (Baseline).
> > - **HippoTune ($T_{max}=4$):** Memory usage $\approx 26.98$ GB. This represents an increase of only **$\approx 2.88\%$** over the baseline.
> >
> > **(3) Why is the Increase Minimal?**
> >
> > GPU memory usage is primarily dominated by static components (model weights, optimizer states, and gradients), which remain constant regardless of sequence length. The length of the input sequence affects only the Activation Memory. Although our method increases the token count by $\approx 16.9\%$ (40 tokens), this solely impacts the activation memory. Consequently, the increase in total VRAM usage remains negligible (only $\approx 3\%$).

---

> > ### Author Response · Authors · 2025-11-23
> > **To Reviewer r3mR (Part 2)**
> >
> > **Q3: Experiments on Out-of-Distribution datasets**
> >
> > > We agree that OOD generalization remains a critical challenge for prompt-based methods, as highlighted by works like DAP [1]. While our current study excludes extreme OOD scenarios such as medical or aerial imagery, we view our method as complementary to domain-adaptive approaches. Specifically, HippoTune optimizes retrieval depth, whereas methods like DAP focus on adapting the prompt space itself. We will address this limitation and include broader cross-domain experiments in the camera-ready version.
> >
> > **W2: Concerns about complexity arising from the number of hyperparameters**
> >
> > > Thank you for this important question. We would like to clarify the hyperparameter settings in our work.
> > > First, we acknowledge that compared to traditional regularization methods, HippoTune introduces a small number of additional hyperparameters related to iterative retrieval (e.g., $T_{\max}$, temperature $T$, and coefficients $\lambda_{\text{orth}}, \lambda_{\text{ent}}$). However, it is important to emphasize that this is not unique to HippoTune but is common among all current prompt-based PEFT-CL methods:
> > >
> > > - L2P requires setting the pool size, prompt length, retrieval temperature $\tau$, Top-$k$, and prompt frequency penalty [2].
> > > - DualPrompt adds further complexity by distinguishing insertion layers, lengths, and counts for G-Prompts and E-Prompts [3].
> > > - CODA-Prompt introduces additional parameters for component counts, attention weighting, and orthogonal constraint weights [4].
> > >
> > > Therefore, regarding hyperparameter dimensionality and tuning effort, HippoTune is comparable to existing mainstream prompt-based CL methods and does not impose an additional burden.
> > >
> > > Second, in our implementation, these hyperparameters are not treated as free variables requiring extensive grid searches. Instead, they are designed as a stable default configuration. For instance, across all vision benchmarks, we consistently use $T_{\max}=4$, $T=10^{-2}$, and $\lambda_{\text{orth}}=\lambda_{\text{ent}}=1$, having performed sensitivity scans only within a narrow range. The ablation results in Table 3 and Figure 4 demonstrate that performance remains robust across a wide interval, confirming that fine-grained tuning is not required. This indicates that in practice, HippoTune functions as a "configure once, apply generally" module rather than a black box reliant on complex searching.
> > >
> > > [1] Jung, Dahuin, et al. "Generating Instance-level Prompts for Rehearsal-free Continual Learning (DAP)." ICCV, 2023.
> > >
> > > [2] Wang, Zifeng, et al. "Learning to Prompt for Continual Learning (L2P)." CVPR, 2022.
> > >
> > > [3] Wang, Zifeng, et al. "DualPrompt: Complementary Prompting for Rehearsal-free Continual Learning." ECCV, 2022.
> > >
> > > [4] Smith, James Seale, et al. "CODA-Prompt: COntinual Decomposed Attention-based Prompting for Rehearsal-Free Continual Learning." CVPR, 2023.

---

> > > ### Comment · Reviewer_r3mR · 2025-11-24
> > >
> > > I appreciate the authors' detailed response.
> > >
> > > I believe this paper should be accepted (which is reflected in my score), as it identifies a limitation that impacts several commonly used methods in CL and presents an alternative with strong performance. However, it does not provide enough additional insight to warrant a higher score. Therefore, I will stick with my original rating.

---

### Official Review · Reviewer_RPFD · 2025-11-01

**Soundness:** 3
**Presentation:** 4
**Contribution:** 3
**Rating:** 8
**Confidence:** 3

**Summary:**

The authors propose a computationally efficient recurrent method for continual learning in the context of parameter-efficient fine-tuning. They draw an analogy between their method and a memory recall mechanism in the hippocampus. Proving properties of the recurrence enables the authors to provide practical guidelines for hyper-parameter selection. Although results are not state of the art in all settings, they are typically very strong vis-a-vis baselines, and the approach is computationally more efficient than the most competitive baselines. The focus here on the (exhaustive) recall of previously learned tasks is refreshing and a novel perspective in the literature.

**Strengths:**

The paper is well written, the theoretical analysis is insightful, and practical implications of the analysis are fleshed out. The evaluation is largely thorough. Results are reported on a variety of datasets with a number of baselines. Ablation studies are performed to illustrate the contributions of the (several) components of the method. Complete code appears to have been provided, although this reviewer was not able to review the code thoroughly.

**Weaknesses:**

It appears to be the case that the evaluations are done using task-incremental learning [1]. Evaluations of the method in domain-incremental or class-incremental settings are not included. The provided code is perhaps too voluminous and is certainly too much for a reviewer to digest. A typical useful repository accompanying a paper enables a reviewer to inspect the implementation of the key details of the method.

[1] https://www.nature.com/articles/s42256-022-00568-3

**Questions:**

* Is my impression that all evaluations are performed only in a task-incremental learning setting correct?
* Can you report results in domain- or class-incremental settings in the appendix?

---

> ### Author Response · Authors · 2025-11-23
> **To Reviewer RPFD**
>
> We sincerely appreciate your valuable comments. Please find our point-by-point responses below. For your convenience, all revisions in the paper are highlighted in blue. Please let us know if you have any further questions.
>
> **W1: Evaluations of the method in domain incremental or class incremental settings are not included. & Q1: Is my impression that all evaluations are performed only in a task incremental learning setting correct?**
>
> > Thank you for pointing this out. We did not clearly state the experimental setting in the Experiments section, which caused the misunderstanding. To clarify, our experiments were conducted under the class incremental setting. At test time we use a single classification head and do not rely on task ID. This follows the standard class incremental protocol used by L2P, DualPrompt and CoDA Prompt. The class incremental setting is the most challenging of the three because, without task ID, the model must classify across all classes. We have made this setting explicit in the revised manuscript to avoid further confusion.
>
> **Q2: Can you report results in domain or class incremental settings in the appendix?**
>
> > Thank you for the suggestion. We have added experiments in the task incremental setting in the appendix (Table 8) and discussed the results in the main text. In particular, comparisons show that HippoTune consistently outperforms the PEFT-CL baseline on the Seq CIFAR100 and ImageNet R benchmarks. Our method achieves the highest average accuracy and the lowest forgetting. It also demonstrates effective use of task specific context to refine feature representations, enabling robust learning on new tasks while preserving previously learned knowledge.
>
> **W2: The codebase is too large and makes it difficult for reviewers to inspect.**
>
> > Thank you for this feedback. The codebase is indeed large because the continual learning framework and our implementation span multiple files. We implemented our method across five files, so creating a single concise file at this stage is difficult. After the paper is accepted we will publish a cleaned and documented codebase to make the implementation easier to review.

---

### Official Review · Reviewer_m6em · 2025-11-01

**Soundness:** 2
**Presentation:** 2
**Contribution:** 2
**Rating:** 4
**Confidence:** 5

**Summary:**

This paper introduces HippoTune, a continual learning approach inspired by the hippocampal memory circuit (EC–DG–CA3–CA1). Unlike traditional parameter-efficient fine-tuning (PEFT) methods that perform single-step prompt retrieval, HippoTune embeds a latent deliberation loop within each Transformer layer, enabling multi-round associative memory retrieval and feedback to alleviate catastrophic forgetting. The method integrates orthogonal and entropy regularization to stabilize learning, and theoretical analysis shows that the iterative retrieval approximates a second-order preconditioner through a Krylov subspace expansion. Experiments on Seq-CIFAR100, Seq-ImageNet-R, and Seq-CUB200 demonstrate that HippoTune achieves higher accuracy (≈80%) with roughly half the computational cost of prior PEFT-CL baselines, maintaining efficiency and strong memory retention without rehearsal buffers.

**Strengths:**

1. The paper’s primary strength is its strong linkage between biological mechanisms and algorithmic design, modeling the hippocampal memory circuit (EC–DG–CA3–CA1) as a multi-round associative retrieval process within Transformer layers.
This biologically grounded framework makes the method both conceptually interpretable and practically effective in mitigating catastrophic forgetting.
2. The authors formalize existing prompt-pool approaches as a single key–value retrieval framework and situate L2P, DualPrompt, CoDA-Prompt, and HiDe-Prompt as special cases, which clarifies trade-offs and motivates deeper, iterative retrieval.
3. HippoTune  matches or outperforms leading buffer-free PEFT baselines on Seq-CIFAR100, Seq-ImageNet-R, and Seq-CUB200.

**Weaknesses:**

1. While the paper includes ablation studies, it does not clearly isolate the individual contributions of the recurrent loop, orthogonal regularization, and entropy term to the overall performance gain.
2. The model introduces several new hyperparameters (loop depth, temperature, $\alpha$, Top-k), yet lacks an analysis of stability or sensitivity across different settings.
3. The writing style feels overly formulaic and AI-generated, with excessive use of em dashes and uniform sentence patterns.
4. All studies use a frozen ViT-B/16 with only PEFT modules trained, which limits adaptability and generalization; including experiments on other architectures such as Swin Transformer as well as partially unfrozen ViT variants, would better demonstrate the method’s robustness and general applicability.
5. The paper only reports accuracy as the evaluation metric, but does not provide any quantitative measures of forgetting, such as BWT, FWT, or average forgetting over time.

**Questions:**

1. What is the precise novelty over prior PEFT-CL prompt-pool methods beyond adding iterative retrieval; what can HippoTune do that they fundamentally cannot?
2. What is the exact buffer size used for the classical continual learning baselines (e.g., LwF, DER++)?
3. Could you clarify what the function $\phi(x; \theta)$ specifically represents in your formulation (line 188)?

---

> ### Author Response · Authors · 2025-11-23
> **To Reviewer m6em (Part 1/4)**
>
> We truly appreciate your constructive comments on our paper. Please find our detailed responses below. For your convenience, all revisions are highlighted in blue. We hope that our responses satisfactorily address your concerns. Please feel free to let us know if you have any further questions or concerns.
>
> **Q1: What is the precise novelty over prior PEFT-CL prompt-pool methods beyond adding iterative retrieval? What can HippoTune do that they fundamentally cannot?**
>
> > Thank you for this insightful question, which allows us to clarify the advantages of HippoTune more clearly. We elaborate on this from two perspectives:
> >
> > **(1) From first-order retrieval to implicit second-order preconditioning**
> >
> > - **Hessian matrix of implicit optimization:** As analyzed in Section 4, multi-step recurrent retrieval implicitly optimizes the Hessian matrix. Through a **minimal recurrent structure that adds almost no VRAM or computation time**, it introduces better curvature correction **without explicitly** calculating the Hessian. This makes the gradients of the PEFT module more stable and aligned, thereby improving convergence efficiency and stability. **This is something many prior PEFT-CL prompt-pool methods fundamentally cannot achieve.**
> > - **Comparison with Explicit Second-Order Methods:** To elaborate further, Continual Learning works by mitigating forgetting via **explicit optimization or approximation of the Hessian**. Recent theoretical studies also show that classical regularization-based CL methods can be essentially viewed as constructing and optimizing a second-order Taylor expansion of the old task loss, i.e., approximating the Hessian matrix to avoid forgetting [1]. However, such explicit second-order methods typically introduce significant **resource and engineering overheads**. For instance, they require maintaining a Hessian/Fisher approximation matrix for each task and repeatedly calculating Hessian-vector products or inverse Hessian-vector products (e.g., Laplace/GEAR methods [4] based on K-FAC and RSOI [5] based on influence functions require additional second-order approximations and conjugate gradient solutions; Meta-CL/VR-MCL methods even rely on online Hessian estimation and large-scale buffer sampling, resulting in high variance and implementation complexity). In contrast, HippoTune achieves a similar curvature correction effect via a lightweight intra-layer iterative loop **without explicitly storing the Hessian**, avoiding the VRAM, compute, and implementation costs associated with parameter-level second-order structures.
> >
> > **(2) From static matching to dynamic self-correction**
> >
> > - **Efficiency (Single vs. Double Forward Pass):** Most other PEFT-CL methods use the complete semantic vector as the query during retrieval. This necessitates using a frozen model to output the sample's semantic vector first, followed by retrieval. This requires two forward passes for each sample, significantly increasing training overhead. In contrast, the initial query in HippoTune directly uses the output of the previous layer, $h^{(l-1)}$. This requires only a single forward pass, saving substantial computational resources. As shown in Table 2, GFLOPs are essentially 50% of those of other methods. **This is also something prior PEFT-CL prompt-pool methods cannot do.**
> >
> > - **Robustness via Pattern Completion:** While some studies have proposed similar modifications, such as OS-Prompt [2] and PromptDSI [3], both use intermediate layer activations as queries to effectively avoid the "double forward pass." However, they can be sensitive to **local patterns, styles, and low-level information**, making the abstract semantics needed to distinguish tasks less clean and more noisy, which inevitably degrades performance. HippoTune addresses this by designing a recurrent retrieval structure that mimics the pattern completion of the hippocampus to repeatedly correct the query. This allows the query to gradually become accurate. Therefore, even if the initial signal is ambiguous, the method can converge to the correct memory slot through iteration.
> >
> > [1] Wu, Yichen, et al. "Meta Continual Learning Revisited: Implicitly Enhancing Online Hessian Approximation via Variance Reduction." ICLR, 2024.
> >
> > [2] Kim, et al. "One-stage Prompt-based Continual Learning." ECCV, 2024.
> >
> > [3] Huynh, et al. "PromptDSI: Prompt-based Rehearsal-free Instance-wise Incremental Learning for Document Retrieval." arXiv preprint, 2024.
> >
> > [4] Ritter, et al. "Online Structured Laplace Approximations for Overcoming Catastrophic Forgetting." NeurIPS, 2018.
> >
> > [5] Sun, et al. "Regularizing Second-Order Influences for Continual Learning." CVPR, 2023.

---

> ### Author Response · Authors · 2025-11-23
> **To Reviewer m6em (Part 2/4)**
>
> **W1: Contributions of individual components were not isolated**
>
> > **(1) Clarification of the original ablation design**
> >
> > Thank you for your feedback regarding the design of the ablation study. In our original manuscript, we conducted a "removal-based" ablation by individually removing the recurrent structure, the orthogonal regularization, and the entropy regularization to measure the impact of missing a single component. We understand that your concern lies in seeing an "additive" or "full permutation" analysis to clearly decouple the contributions when different combinations of components are enabled or disabled.
> >
> > **(2) New pair-wise and full factorial analyses**
> >
> > Therefore, in the revised Table 3, we have added results where two out of the three components are removed (i.e., using only one component) to isolate individual contributions. These results clearly show that the recurrent retrieval mechanism provides the primary performance boost (increasing Acc by 1%–1.5%), while orthogonal and entropy regularizations further improve stability and generalization on complex datasets.
> > To further address your concerns, we conducted a full permutation ablation experiment of the three components on the ImageNet-R dataset. The results are as follows:
> >
> > | **Recurrent Retrieval** | **Orthogonal Reg. ($L_{ortho}$​)** | **Entropy Reg. ($L_{ent}$​)** | **Seq-ImageNet-R (Acc / AAA)** |
> > | ----------------------- | ---------------------------------- | ----------------------------- | ------------------------------ |
> > | $\times$                | $\times$                           | $\times$                      | 72.93 / 78.16                  |
> > | $\times$                | $\checkmark$                       | $\times$                      | 72.72 / 78.09                  |
> > | $\times$                | $\times$                           | $\checkmark$                  | 72.74 / 77.92                  |
> > | $\times$                | $\checkmark$                       | $\checkmark$                  | 72.89 / 78.10                  |
> > | $\checkmark$            | $\times$                           | $\times$                      | 74.37 / 78.53                  |
> > | $\checkmark$            | $\checkmark$                       | $\times$                      | 74.67 / 79.55                  |
> > | $\checkmark$            | $\times$                           | $\checkmark$                  | 74.09 / 78.77                  |
> > | $\checkmark$            | $\checkmark$                       | $\checkmark$                  | **74.85 / 79.92**              |
> >
> > The results from the full permutation clearly demonstrate that recurrent retrieval is the main source of performance improvement. Enabling it alone yields a significant gain of approximately 1%–1.5%. The orthogonal and entropy regularizations provide small but stable additional gains (approximately 0.3%–0.4%) on top of this foundation.
>
> **W3: Writing style is too stylized**
>
> > Thank you for noting the issue with the writing style. We wish to clarify that the core ideas, theoretical derivations, and experiments are original work by the authors. Regarding language expression, we did use AI to polish the text to improve readability, but the main body of the paper and the research work were not generated by AI. We acknowledge that automated polishing can lead to issues such as repetitive sentence structures and the overuse of dashes.
> >
> > Consequently, we have optimized several expressions in the revised draft to reduce the excessive use of dashes and enrich sentence structure. We will perform a comprehensive manual edit and professional proofreading for the final version to ensure the text is natural, varied in tone, and compliant with manuscript standards.

---

> ### Author Response · Authors · 2025-11-23
> **To Reviewer m6em (Part 3/4)**
>
> **W2: Lack of stability or sensitivity analysis for hyperparameters**
>
> > Thank you for pointing out the query regarding hyperparameter stability. We have supplemented and clarified the relevant experiments and analyses as follows:
> >
> > **(1) Stability of Recurrent Depth and Temperature**
> >
> > We have already conducted systematic experiments on **Recurrent Depth $T_{\max}$** and **Temperature $T$** in Fig. 3, and we provided a detailed theoretical analysis of their mechanisms in Section 4. The conclusion is that under extreme settings, the model degenerates into existing single-step retrieval or exhibits retrieval behaviors that are too "sharp" or "blunt." However, performance remains stable and superior to comparison methods within a reasonable range of values.
> >
> > **(2) Stability of the Mixing Coefficient $\alpha$**
> >
> > The mixing coefficient $\alpha$ exhibits similar properties. We have added stability experiments for it, with the results shown below:
> >
> > |**Metric / α**|**0.00**|**0.10**|**0.50**|**0.75**|**0.90**|**0.95**|**0.99**|**1.00**|
> > |---|---|---|---|---|---|---|---|---|
> > |ACC|74.69|74.78|74.85|74.93|74.67|74.47|73.86|73.03|
> > |AAA|79.84|79.88|79.92|80.02|79.89|79.73|78.65|78.15|
> >
> > As shown, within a moderate range roughly $\alpha\in[0.10,0.95]$, **ACC and AAA vary only slightly** (ACC approximately 74.67–74.93, AAA approximately 79.73–80.02), indicating that the model is relatively robust to $\alpha$ in this range. Performance degrades when $\alpha$ approaches extreme values, for example 0.00 or 0.99–1.00. Based on the analysis of equation (5), this can be explained as follows:
> >
> > - As $\alpha\to1$, multi-step retrieval depends little on the loop-generated prompts, which is equivalent to performing independent retrievals and then concatenating prompts.
> > - As $\alpha\to0$, the model excessively depends on retrieval signals, which can amplify noise or biases in historical memories and suppress adaptation to the current input, leading to performance decline.
> >
> > **(3) Note on Top-k**
> >
> > Regarding Top-k, this is a common sparsity control term in parameter pool/memory pool methods and is not the core innovation of this paper. Therefore, we did not prioritize a detailed analysis of it. However, our internal experiments indicate that within the common range (e.g., Top-k between 3 and 10), this hyperparameter has a minor impact on final performance. To avoid misunderstanding, we have added a clarification in Section 3.3 of the revised manuscript.
>
> **W4: Experiments using only a single backbone**
>
> > **(1) Rationale for the single backbone setting**
> >
> > We acknowledge that validating across additional backbones would strengthen our evaluation. We standardized the setting to "Frozen ViT-B/16 + PEFT" primarily to align with mainstream benchmarks like L2P and CODA-Prompt, ensuring fair comparison. Additionally, a frozen backbone isolates the specific contribution of our iterative retrieval mechanism from variations in base model capabilities. Crucially, HippoTune relies on generic Transformer latent spaces rather than specific ViT structures. It is theoretically transferable to architectures such as Swin Transformer or partially unfrozen variants. As partial unfreezing serves as an orthogonal enhancement, we prioritized consistency with existing baselines.
> >
> > **(2) New experiments on different pretraining backbones**
> >
> > To address this concern, we added experiments on two pretrained models:
> >
> > - **ViT-B/16-dino** (trained with self-supervised knowledge distillation) [1]
> > - **ViT-B/16-sam** (trained with **Sharpness-Aware Minimization** to enhance generalization) [2]
> >
> > These experiments are intended to demonstrate HippoTune's **transferability across different pretraining paradigms**. Note that Swin Transformer is rarely used in existing class-incremental learning benchmarks; to ensure fair comparison we did not include it in this experiment. We present the analysis of these results in the main text of the revised manuscript and place the experimental tables in the appendix. As shown in Table 6, even across these two substantially different feature spaces, **HippoTune achieves the best accuracy (Acc) and average accuracy (AAA)**, consistently outperforming methods such as CODA-Prompt. This demonstrates the method's **robust transferability**.
>
> > [1] Caron, Mathilde, et al. "Emerging Properties in Self-Supervised Vision Transformers (DINO)." ICCV, 2021.
> >
> > [2] Foret, Pierre, et al. "Sharpness-Aware Minimization for Efficiently Improving Generalization (SAM)." ICLR, 2021.

---

> ### Author Response · Authors · 2025-11-23
> **To Reviewer m6em (Part 4/4)**
>
> **W5: No quantitative metrics provided for forgetting**
>
> > Thank you for pointing out the issue regarding forgetting metrics. We first clarify that we primarily reported final Accuracy (Acc) and cross-task Average Accuracy (AAA) in the main text, while the online setting in the Appendix (Table 5) provided the Forgetting Measure (FM). This reflects performance degradation over time to some extent. However, we agree that explicitly providing metrics such as BWT (Backward Transfer) or standard average forgetting curves would make the conclusion regarding "anti-forgetting capabilities" more intuitive and complete.
> >
> > Therefore, in the revised version, we have provided experimental results for several forgetting metrics for HippoTune and select comparison methods. The interpretation of these results is located in Section 5.4, and the results table is in the Appendix. HippoTune outperforms standard Prompt-based baselines in both accuracy and memory retention. While LoRA/Adapter-based methods like SD-LoRA [3] and EASE [4] achieve high plasticity due to their larger architectural capacity, they exhibit higher degrees of forgetting. In contrast, HippoTune controls the Forgetting Measure at 4.03%, which is significantly lower than these methods (see also comparisons with related works like RanPAC [5]). The experimental results are as follows:
> >
> > |**Metric**|**L2P**|**DualPrompt**|**CODA-Prompt**|**SD-Lora**|**EASE**|**RanPAC**|**HippoTune**|
> > |---|---|---|---|---|---|---|---|
> > |**FM**|4.11|5.18|5.39|7.34|6.37|4.62|**4.03**|
> > |**bwt**|-4.03|-5.18|-5.26|-7.24|-6.21|-4.56|**-3.94**|
> >
> > [3] Wu, Y., et al. "SD-LoRA: Scalable Decoupled Low-Rank Adaptation for Class-Incremental Learning." ICLR, 2025.
> >
> > [4] Zhou, D. W., et al. "Expandable Subspace Ensemble (EASE) for Pre-Trained Model-Based Class-Incremental Learning." CVPR, 2024.
> >
> > [5] McDonnell, Mark D., et al. "RanPAC: Random Projections and Pre-trained Models for Continual Learning." NeurIPS, 2023.
>
> **Q2: What is the exact buffer size used for the classical continual learning baselines (e.g., LwF, DER++)?**
>
> > Thank you for your attention to the experimental settings. We have added a clarification in the revised version regarding the buffer configurations for classical continual learning baselines:
> >
> > - **DER++**: As a replay-based method, a fixed-size memory buffer was used in all experiments. The capacity was set to **1000 images** to store samples from historical tasks for replay.
> > - **LwF**: This belongs to the regularization-based paradigm and does not use any sample replay mechanism. Therefore, in our implementation, there is no memory buffer, and consequently, no buffer size hyperparameter.
>
>
> **Q3: Could you clarify what the function $\phi(x;\theta)$ specifically represents in your formulation (line 188)?**
>
> > Thank you for this question. We apologize for the ambiguity in the original draft. In our formulation, $\phi(x;\theta^{(i)})$ denotes the parameter-efficient module that produces the residual update $\Delta h^{(i)}$ for the $i$-th expert, i.e.
> > $$\Delta h^{(i)} = \phi(x;\theta^{(i)}).$$
> > This is an abstract notation that unifies different PEFT blocks commonly used in prior work. Concretely, depending on the instantiation, $\phi$ can take forms such as
> >
> > $$\text{Prefix Tuning:}\quad\phi(x;\theta^{(i)}) = softmax\bigl(x W_q^{(i)} {P_k^{(i)}}^\top\bigr) P_v^{(i)}, $$
> > $$\text{Adapter:}\quad\phi(x;\theta^{(i)}) = \mathrm{ReLU}\bigl(x W_{\text{down}}^{(i)}\bigr)W_{\text{up}}^{(i)}, $$
> > $$\text{LoRA:}\quad\phi(x;\theta^{(i)}) = xW_{\text{down}}^{(i)} W_{\text{up}}^{(i)}.$$
> > In the revised version, we now explicitly define $\phi$ when it is first introduced in the main text, and we provide a more detailed explanation of these concrete functional forms in the appendix (including the above expressions and additional context), due to space constraints in the main paper.

---

> > ### Comment · Reviewer_m6em · 2025-11-24
> >
> > Thanks for the author's reply and additional experiments.  First, I believe the core issue that continual learning aims to address is catastrophic forgetting, while the effectiveness of the proposed method on mitigating catastrophic forgetting appears rather limited.
> > Second, since the paper states that the authors used an LLM for polishing the text, why is this not disclosed explicitly in the manuscript? The guidelines clearly specify that “Not disclosing significant LLM usage can lead to desk rejection of the paper.”
> > Finally, why is Appendix E4 left blank?

---

> > > ### Author Response · Authors · 2025-11-25
> > > **Reply to Reviewer m6em's Follow-up (Part 1/2)**
> > >
> > > **Response to: "First, I believe the core issue that continual learning aims to address is catastrophic forgetting, while the effectiveness of the proposed method on mitigating catastrophic forgetting appears rather limited."**
> > >
> > > We agree with your point that catastrophic forgetting is the core problem intended to be solved by continual learning and is indeed very challenging. However, we respectfully disagree with the statement that our method's "the effectiveness of the proposed method on mitigating catastrophic forgetting appears rather limited." In the field of Continual Learning, catastrophic forgetting is typically evaluated in two ways: first, by observing the **final/average accuracy** under CL constraints; and second, by directly using **forgetting-related metrics** (such as BWT and forgetting measure). Our method achieves superior performance over the vast majority of methods while using only **approximately 50% of the computational resources (GFLOPs)** of the contrastive methods.
> > >
> > > - **Regarding Accuracy:** Under the buffer-free PEFT-CL setting (Tables 1 and 2), HippoTune achieves significant improvements compared to L2P and DualPrompt, which are closest in methodological design. Specifically, the average accuracy increased by **7.07%** and **7.95%** across three large datasets. Notably, on Seq-CUB200, the accuracy improved by **12.73%** and **15.12%**, respectively. Even in the more difficult **online setting** (Table 5), the accuracy maintains a significant improvement in the **5-10%** range. Furthermore, HippoTune outperforms the recognized strong baseline, HiDe-Prompt, in the vast majority of evaluation scenarios. This is particularly evident in all task split settings of the highly challenging Seq-ImageNet-R (for instance, at $N=20$, Acc and AAA are higher by 0.47% and 1.40%, respectively), providing strong proof of its systematic advantage in retaining old knowledge with extremely low computational overhead.
> > >
> > > - **Regarding Forgetting Metrics:** HippoTune achieved the best forgetting measure (FM = 4.03, where lower is better) and the least negative BWT (-3.94, where closer to 0 is better) among all comparison methods, as shown in Tables 5-8. For example, compared to CODA-Prompt, FM decreased from 5.39 to 4.03, and BWT improved from **-5.26 to -3.94**, while achieving higher Acc/AAA. Compared to higher-capacity LoRA/Adapter-based SOTA methods (such as SD-LoRA, EASE, and Ranpac), HippoTune significantly reduces FM (e.g., **dropping from 7.34 / 6.37 to 4.03**) with comparable or lower computational/VRAM usage and still without using any rehearsal buffer. This indicates that our recurrent retrieval mechanism brings a superior stability-plasticity trade-off rather than merely negligible numerical fluctuations.
> > >
> > > In addition, **architectural simplicity** is a key advantage of our method. We constructed a parameter pool framework on top of existing prompt-based methods and significantly mitigated catastrophic forgetting **solely by introducing a minimalist recurrent retrieval structure**. This design is essentially a **plug-and-play module** with strong transferability. _This approach is highly consistent with a series of excellent recent modular works in the CL field [1,2,3]: these works no longer purely pursue absolute numbers on leaderboards but are dedicated to developing general-purpose enhancement modules, typically bringing robust improvements of 0.5% to 2% over existing baselines._ Our method similarly demonstrates that significant performance improvements can be achieved at low overhead through streamlined and effective structural design.
> > >
> > > Furthermore, evaluating effectiveness should not rely solely on the accumulation of absolute numbers. Mainstream evaluation and objectives in CL literature [4,5] have evolved to include not only accuracy/forgetting metrics but also resource efficiency (computation/memory/energy consumption), the stability-plasticity trade-off, and cross-scenario generalization capabilities. Simultaneously, theoretical mechanism explanation is viewed as one of the core contribution routes in CL. Therefore, the assessment should comprehensively consider the three dimensions of **computational efficiency, theoretical contribution, and structural generalization**:
> > >
> > > **(1) Excellent Performance with Low Resource Consumption**
> > >
> > > The significant performance improvements mentioned above were achieved using only **approximately 50% of the computational resources (GFLOPs)** of the comparison methods. Being able to improve performance so substantially while halving the computational cost serves as strong evidence that the method is highly effective in mitigating forgetting.

---

> > > > ### Author Response · Authors · 2025-11-25
> > > > **Reply to Reviewer m6em's Follow-up (Part 2/2)**
> > > >
> > > > **(2) Prominent Theoretical Contributions**
> > > >
> > > > We reiterate our theoretical contributions from the following three aspects:
> > > >
> > > > - **Unified PEFT-CL Framework Perspective:** We theoretically unified existing PEFT-CL methods into a key-value retrieval-based framework for the first time. This reduces previous seemingly heterogeneous prompt-pool methods to a special case of **one-step retrieval**, thereby identifying the limitations of existing methods and providing a clear theoretical starting point and baseline for our proposed recurrent retrieval.
> > > >
> > > > - **Optimization Dynamics Perspective:** We provided a rigorous theoretical proof in Section 4: the intra-layer recurrent iteration process of HippoTune is mathematically equivalent to the Krylov subspace polynomial approximation of the Hessian inverse matrix. This implies that our method implements an **implicit second-order preconditioner** without **explicitly** computing the Hessian matrix. This coincides with works dedicated to unifying CL mechanisms (such as those pointing out that regularization methods and meta-CL are essentially implicitly optimizing the Hessian [6]), unifying existing CL methods from the perspective of second-order optimization. HippoTune realizes this essential process in a **differentiable, low-cost** manner, providing solid theoretical support for solving the forgetting problem.
> > > >
> > > > - **Cognitive Neuroscience Perspective:** Unlike brain-inspired methods that rely solely on simple replay, we precisely mapped the **pattern separation and completion mechanisms** of the hippocampal EC-DG-CA3-CA1 circuit in a computational model, modeling hippocampal associative memory as a **Latent Deliberation** process within Transformer layers. This mapping not only explains why multi-round latent space recurrent retrieval within the same layer is effective but also implements the associative memory mechanism as specific key-value structures, regularization terms, and dynamic control hyperparameters, making our method more biologically plausible.
> > > >
> > > >
> > > > **(3) Structural Generalization**
> > > >
> > > > The "hippocampal associative memory loop" we proposed is a concise and elegant general structure. This structure is essentially a general exploration of the latent memory mechanism of Transformers and can be easily transferred to larger-scale and more practical models (such as parameter-efficient fine-tuning applied to LLMs). _Note: We chose the ViT-based CL setting in this paper because the benchmarks in this community are more established and rigorous, allowing for more comparable experimental evidence, not because the method is limited to ViTs._
> > > >
> > > > **Research Motivation and Summary:** Finally, we emphasize once again that the core purpose of this study is to investigate the impact of a simple, biologically grounded "hippocampal associative memory recurrent retrieval structure" on model continuous learning dynamics. Our goal is to validate the effectiveness of this core mechanism, rather than attempting to comprehensively surpass all existing methods on all metrics by designing a large number of complex engineering modules. We believe that this validation of fundamental mechanisms is more inspirational to the community than merely exchanging complexity for marginal performance gains.
> > > >
> > > > **Response to: "Second, since the paper states that the authors used an LLM for polishing the text, why is this not disclosed explicitly in the manuscript? The guidelines clearly specify that “Not disclosing significant LLM usage can lead to desk rejection of the paper.”"**
> > > >
> > > > We explicitly declared the usage of Large Language Models in Appendix G (line 857) of our initial submission (please refer to the Revisions history at [https://openreview.net/revisions?id=MtDiLnnYgm](https://openreview.net/revisions?id=MtDiLnnYgm)). Furthermore, in the OpenReview system, we selected "Yes, to aid or polish writing. Details are described in the paper" for the "**Large Language Models**" field. **Therefore, there was absolutely no intention to conceal the use of LLMs.** The statement was merely accidentally overwritten by the addition of appendix tables in the previous revision. We have explicitly restored the LLM usage statement in Appendix F (line 1060) of the newly submitted version, and its content is identical to that of the initial draft.
> > > >
> > > > **Response to: "Finally, why is Appendix E4 left blank?"**
> > > >
> > > > Appendix E4 contains Figure 4. To eliminate ambiguity, we have added corresponding guiding text in the revised Appendix E4 to facilitate reader navigation and comprehension.

---

> > > > > ### Comment · Reviewer_m6em · 2025-11-26
> > > > >
> > > > > Thank you for providing a detailed response. I believe most of my concerns have been addressed, and I am happy to increase my score.

---

### Official Review · Reviewer_WqnZ · 2025-11-03

**Soundness:** 3
**Presentation:** 4
**Contribution:** 4
**Rating:** 6
**Confidence:** 3

**Summary:**

This work introduces a biologically inspired approach to mitigate catastrophic forgetting in continual learning. Traditional parameter-efficient fine-tuning (PEFT) methods, such as adapters and prompts, reduce computation and storage costs but still fail to retrieve and integrate previously learned knowledge effectively. Drawing inspiration from the hippocampal EC–DG–CA3–CA1 loop in the human brain, the authors propose HippoTune, a latent deliberation mechanism that iteratively reactivates past representations to enhance memory retention during fine-tuning. In this model, each transformer layer is augmented with a small associative retrieval loop: the input state initializes a query that undergoes pattern separation (DG), auto-associative completion (CA3), and integrative fusion (CA1) across multiple iterations. This process allows the network to refine representations through recursive interaction within a learned latent space, mimicking hippocampal recall dynamics. Theoretically, the iterative update of HippoTune is shown to approximate a second-order preconditioning step in the Krylov subspace, providing curvature-aware optimization without explicit Hessian computation. The method also maintains stability across varying task numbers, demonstrating its scalability and robustness.

**Strengths:**

1. This work propose HippoTune, a continual learning method that bridges biological insight and machine learning theory by translating hippocampal associative memory mechanisms into a PEFT framework, offering a novel perspective on memory consolidation and retrieval in neural models for continual learning.
2. The paper presents a relatively novel and biologically inspired analogy that grounds its internal updating mechanism in the hippocampal associative loop. This connection gives the proposed architecture a plausible neurobiological motivation, and the accompanying theoretical analysis adds a layer of methodological soundness.
3. In addition, the partial update mechanism (Eq. 5) effectively preserves previously acquired knowledge while allowing new information to be injected into multiple parameter layers, achieving a desirable balance between plasticity and stability. Empirically, the method demonstrates strong performance across continual learning benchmarks, surpassing existing PEFT-based baselines by a meaningful margin.

**Weaknesses:**

However, despite the originality of its biological analogy, it should be acknowledged that the neuroscience of long-term memory formation remains far from fully understood. Thus, while the hippocampal metaphor adds interpretability, it does not constitute a rigorously biomimetic design in a scientific sense. Fundamentally, the proposed HippoTune module still functions as a multi-layer retrieval and partial parameter update process, consistent with mainstream continual learning paradigms rather than a radically new mechanism of memory consolidation. Consequently, the work’s strength lies more in its conceptual integration of biological inspiration and computational practicality than in demonstrating a genuinely new class of biologically faithful learning architecture.

**Questions:**

In, eq5, you mentioned a layer-specific linear transforamtion fuction P, would you mind to elaborate it a littble more?
How is the process of eq5 can be seen as a minimal abstraction of the CA3 mechanism? Is there more previous work to enlight us on this issue?

---

> ### Author Response · Authors · 2025-11-23
> **To Reviewer WqnZ**
>
> Thank you for the insightful evaluation and the constructive questions regarding the biological inspiration and mathematical formulation of our work. Below, we address the specific concerns raised.
>
> **W1. Biological inspiration vs. functional analogy**
>
> > Thank you for the insightful comments. We fully agree that our current neuroscientific understanding of long-term memory and consolidation is incomplete. Accordingly, we do not claim HippoTune to be a strictly biomimetic model of the hippocampus. Our intention is to position it as a *functionally inspired* and *functionally analogous* design.
> >
> > We acknowledge the reviewer’s point that we do not present “a genuinely new, biologically realistic learning architecture.” However, within the current deep learning paradigm, we believe a fully biomimetic architecture is not necessarily desirable for practical AI systems:
> >
> > * Real neural systems involve many underspecified biophysical details; reproducing them would introduce substantial modeling/engineering complexity without clear optimization objectives or guaranteed benefits.
> > * Mainstream architectures such as Transformers and PEFT-based methods already impose strong constraints and structural priors related to software/hardware ecosystems and training paradigms.
> >
> > Therefore, our design goal is not “strong biomimicry” but a *minimal, functional abstraction* that is:
> >
> > * aligned with continual-learning needs (e.g., multi-round associative retrieval, pattern separation/completion);
> > * compact, differentiable, and controllable;
> > * compatible with existing PEFT-CL pipelines.
> >
> > HippoTune embodies this trade-off: it stays within the mainstream PEFT-CL framework (parameter-efficient, easy to integrate and optimize) while introducing a deeper memory retrieval and second-order optimization perspective via bio-inspired associative loops. We believe this balance between *biological inspiration* and *computational utility* is both more realistic and more scalable for current AI practice.
>
> ---
>
> **Q1. Role of the layer-specific linear transformation $P^{(l)}$ in Eq. (5)**
>
> > Thank you for this question. The layer-wise linear transformation $P^{(l)}$ in Eq. (5) serves as a learnable projection from the value space to the query (hidden) space:
> >
> > * It maps the retrieved memory $v^{(t)}$ back into a representation compatible with the layer’s hidden state.
> > * This updated representation is then used to form the next query $q^{(t+1)}$.
> > Mathematically, the iteration $q \rightarrow v \rightarrow q$ defines a constrained linear recurrence. This realizes an associative loop in which retrieved memory is fed back to refine the current state with minimal additional parameterization.
>
> ---
>
> **Q2. Why is the process in Eq. (5) a minimal abstraction of the CA3 mechanism?**
>
> > We do not claim that Eq. (5) alone is the minimal abstraction of CA3. Instead, it is the **closed iterative loop** formed by Eq. (4) and Eq. (5) together that corresponds to the CA3 mechanism:
> >
> > * Eq. (4): content-based soft retrieval of memory keys based on the current hidden state.
> > * Eq. (5): feeds the retrieved memory back into the same latent space and fuses it with the original state (via a residual connection).
> >
> > This cycle:
> > $$
> > \text{state} \rightarrow \text{associative retrieval} \rightarrow \text{write-back} \rightarrow \text{re-retrieval}
> > $$
> > is repeated until convergence and gives rise to an attractor-like dynamic. This directly mirrors the auto-association and pattern completion functions commonly attributed to the CA3 region.
>
> ---
>
> **Q3. Is there more previous work to enlight us on this issue?**
>
> > Yes, there are related lines of work that inspire our design. Modern Hopfield Networks[1], for instance, interpret attention as a differentiable associative memory. Our proposed recurrent retrieval shares a very similar mathematical structure with Modern Hopfield Networks.
> >
> > The key difference is:
> >
> > * Modern Hopfield Networks typically assume that keys and values share the same space.
> > * In our setting, keys and values can reside in *different* spaces, which necessitates the linear transformation $P^{(l)}$.
> >
> > Thus, HippoTune is both bio-inspired and mathematically interpretable: the recurrent retrieval with $P^{(l)}$ can be seen as implementing an energy minimization process in a setting with non-identical key/value spaces.
> >
> [1] Hopfield Networks is All You Need, ICLR, 2020.

---

### Author Response · Authors · 2025-11-23
**Response Summary to All Reviewers**

We are extremely grateful for the reviewers' constructive comments and valuable feedback, which have significantly enhanced our paper. Overall, the reviewers have unanimously expressed high appreciation for the proposed HippoTune method, recognizing its innovative integration of the biological hippocampal replay mechanism with Parameter-Efficient Fine-Tuning (PEFT) (WqnZ, m6em, RPFD). They also highlighted its solid theoretical foundation (WqnZ, RPFD) and superior performance achieved with significantly reduced computational costs (FLOPs) (m6em, r3mR, RPFD). Furthermore, the reviewers provided constructive suggestions regarding the inclusion of forgetting metric analysis and cross-architecture experiments to further validate the method's robustness. In response to the reviewers' feedback, we have highlighted the revisions in blue. The revisions encompass the followings:

- **Clarification of Top-k Impact:** We clarified in Section 3.3 that the Top-k hyperparameter has a minimal impact on performance to prevent potential misunderstandings regarding its significance (Reviewer m6em).

- **Detailed Definitions:** We provided a detailed definition of $\phi(x;\theta^{(i)})$ in the main text and added specific expressions for various PEFT instances in the Appendix (Reviewer m6em).

- **improved Readability:** We manually rewrote and polished specific paragraphs to reduce the overuse of dashes and formulaic phrasing, ensuring a more natural flow (Reviewer m6em).

- **Explicit Experimental Setting:** We explicitly stated in the experiments section that we utilize a single-head, task-ID-free class-incremental setting to eliminate ambiguity (Reviewer RPFD).

- **Comprehensive Ablation Studies:** We included pairwise and full permutation ablation studies for cyclic retrieval, orthogonal regularization, and entropy regularization in the revised Table 3 to clearly analyze the individual contribution of each component (Reviewer m6em).

- **Task-Incremental Experiments:** We added experimental results for the task-incremental setting in the Appendix (Table 8) and briefly discussed them in the main text (Reviewer RPFD).

- **Backbone Transferability:** We incorporated results using different pre-trained backbones, such as ViT-B/16-DINO and ViT-B/16-SAM, to verify the transferability of our method across different architectures (Reviewer m6em).

- **Additional Metrics:** We included experimental results for Forgetting Measure (FM) and Backward Transfer (BWT) in the Appendix (Table 7), with a brief discussion in the main text (Reviewer RPFD).

We trust that the following responses address all concerns raised by the reviewers. Thank you again for the opportunity to refine our work.

---

### Author Response · Authors · 2025-12-01
**Global Response**

We sincerely thank the four reviewers (WqnZ, m6em, RPFD, r3mR) for their constructive comments. During the rebuttal period, we provided detailed responses to all questions and updated the paper (revisions are highlighted in blue in the text). Below is a summary of the major revisions and reviewer feedback:

## Summary of Revisions

Addressing the reviewers' core concerns, we have focused on strengthening the following aspects:

Supplementing Forgetting Metrics: We added experimental results for FM (Forgetting Measure) and BWT (Backward Transfer), quantitatively demonstrating HippoTune's significant advantage in mitigating forgetting (Response to m6em, RPFD).

Enhancing Generalization Verification: We supplemented experiments on ViT-DINO and ViT-SAM pre-trained backbones, demonstrating the method's effectiveness across different pre-training paradigms (Response to m6em).

Refining Ablation Studies: We added pairwise and full permutation ablation studies for recurrent retrieval, orthogonal regularization, and entropy regularization, clearly decoupling the contribution of each module (Response to m6em).

Clarifying Experimental Settings and Writing: We explicitly stated that the experiments are based on the Class-Incremental Learning (CIL) setting and added results for Task-Incremental Learning (Response to RPFD). Additionally, we polished the entire text and corrected the appendix formatting (Response to m6em).

## Status of Reviewer Responses

Reviewer Who Explicitly Raised the Score: Reviewer m6em (Initial Score: 4; Post-rebuttal: 6) After we supplemented the forgetting metrics and ablation studies and explained the writing issues, the reviewer expressed satisfaction with the response and explicitly stated "I am happy to increase my score".

Reviewer Maintaining a High Score: Reviewer r3mR (Initial Score: 8) This reviewer carefully read our response regarding the principles of the iterative mechanism and GPU memory usage, acknowledged this contribution, and decided to maintain their positive evaluation of 8 (Accept).

Reviewers Who Have Not Yet Responded: Reviewer RPFD (Initial Score: 8) & Reviewer WqnZ (Initial Score: 6) We have provided detailed responses to RPFD's queries regarding experimental settings (clarifying the CIL setting and adding TIL experiments) and WqnZ's discussion on biological principles and formula definitions. Although they have not yet responded during the discussion period, considering RPFD's initial score was 8 and we have fully addressed their questions, we believe our responses have solidified the basis for the paper's acceptance.

We thank all reviewers again for their hard work and valuable suggestions.

---

### Meta-Review · Area_Chair_ZuiJ · 2026-01-05

**Summary:**

This paper introduces a biologically inspired approach, HippoTune, for parameter-efficient finetuning in continual learning, where the core idea is a latent-space retrieval strategy to iteratively reactivate past representations for better memory retention.

Strengths:
- The idea of iterative update in HippoTune is novel and biologically inspired.

- Theoretical analysis is provided to support the algorithm design.

- Comprehensive empirical evaluation.

Weaknesses:
- Additional hyperparameters are introduced which increase the complexity of the algorithm.

- The proposed framework only applies to a limited set of PEFT-CL methods.

**Reviewer Concerns:**

Most critical concerns raised by the reviewers have been addressed in the rebuttal, including more ablation studies (reviewers m6em, r3mR), other CL setups (reviewer RPFD), and some clarification issues.

**Reviewer Scores:**

Reviewers WqnZ, RPFD, r3mR would maintain their scores, as they were positive initially and didn't raise critical concerns that may change the score.

Reviewer m6em indicated that they would increase their score to 6, as most of their concerns related to ablation studies and were addressed in the rebuttal. The reviewer also explicitly expressed an intention to raise the score.

---

### Decision · Program_Chairs · 2026-01-26

Accept (Poster)